# The Minimal Phenomenal Experience questionnaire (MPE-92M): Towards a phenomenological profile of "pure awareness" experiences in meditators

Alex Gamma[1], Thomas Metzinger[2]

1 Research Department, University Hospital of Psychiatry, Zürich, Switzerland, 2 Arbeitsbereich Theoretische Philosophie, Philosophisches Seminar, Johannes Gutenberg University, Mainz, Germany

☯ These authors contributed equally to this work.
* alex.gamma@uzh.ch (AG); metzinger@uni-mainz.de (TM)

**Data Availability Statement:** The data, documentation and all language versions of the MPE-92M questionnaire are publicly available as a

## Abstract

### Objective

To develop a fine-grained phenomenological analysis of "pure awareness" experiences in meditators.

### Methods

An online survey in five language versions (German, English, French, Spanish, Italian) collected data from January to March 2020. A total of 92 questionnaire items on a visual analogue scale were submitted to exploratory and confirmatory factor analysis.

### Results

Out of 3627 submitted responses, 1403 were usable. Participants had a median age of 52 years (range: 17–88) and were evenly split between men and women (48.5% vs 50.0%). The majority of meditators practiced regularly (77.3%), were free of diagnosed mental disorders (92.4%) and did not regularly use any psychoactive substances (84.0%). Vipassana (43.9%) followed by Zen (34.9%) were the most frequently practiced meditation techniques. German (63.4%) and English (31.4%) were by far the most frequent questionnaire languages. A solution with 12 factors explaining 44% of the total variance was deemed optimal under joint conceptual and statistical considerations. The factors were named "Time, Effort and Desire," "Peace, Bliss and Silence," "Self-Knowledge, Autonomous Cognizance and Insight," "Wakeful Presence," "Pure Awareness in Dream and Sleep," "Luminosity," "Thoughts and Feelings," "Emptiness and Non-egoic Self-awareness," "Sensory Perception in Body and Space," "Touching World and Self," "Mental Agency," and "Witness Consciousness." This factor structure fit the data moderately well.

### Conclusions

We have previously posited a *phenomenological prototype* for the experience of "pure awareness" as it occurs in the context of meditation practice. Here we offer a tentative 12-

ZIP-file on the Open Science Framework website. URL: https://osf.io/gb76x/download.

**Funding:** This work was funded by a 5-year Fellowship awarded to TM by the Gutenberg Research College, Johannes Gutenberg-Universität Mainz (https://www.grc.uni-mainz.de/). The funder had no role in study design, data collection and analysis, decision to publish, or preparation of the manuscript.

**Competing interests:** The authors have declared that no competing interests exist.

factor model to describe its phenomenal character in a fine-grained way. The current findings are in line with an earlier study extracting semantic constraints for a working definition of minimal phenomenal experience.

## Introduction

The concept of "pure consciousness" or "pure awareness" has a long tradition in the literature on contemplative practices. It refers to the meditator's subjective experience of consciousness *as such*, wherein he or she is non-conceptually aware of *being aware*. Pure awareness is often described as a *contentless* form of experience, and it has played a great role in Eastern philosophical traditions. Over the centuries, contemplative practice has mostly taken place against the background of religious belief systems like Buddhism or Hinduism, with meditators trying to achieve a soteriological goal like "liberation" or "enlightenment." Accordingly, the phenomenological taxonomies of such states have been shaped by metaphysical belief systems and an ancient, traditional cultural context. However, during the last 50 years, a historically new situation has emerged: Millions of practitioners in Western societies meditate regularly, on a daily basis, but many of them do so in a secular context and describe themselves a "spiritual but not religious" (SBNR).

Scientific research in the past century has occasionally taken an interest in states of "pure awareness," but mostly from a descriptive angle and using qualitative methods ([1, 2]; for a review see [3]). Quantitative, including psychometric, approaches have been lacking, particularly with regard to the phenomenology of such experiences [3–6].

What has emerged from the little research conducted so far has been in line with the traditional literature, showing states of pure awareness to be characterized by an "absence of space and time, or body sense" and by the experience of "peacefulness" and "unboundedness" [2]. Some authors have studied particular aspects of pure consciousness experiences, such as non-dual awareness. For example, Hanley et al. [7] found two dimensions of the experience of non-dual awareness: self-transcendence (a dissolving of the boundaries of the self and a feeling "oneness" with everything around) and bliss (feelings of peacefulness, love and blissful warmth). Most topical research, however, is more tangential to the phenomenon of pure awareness, which appears as part of a construct (e.g. in the "mystical" factor of the Mystical Experience Questionnaire MEQ30 [8]), but is not itself the focus of investigation.

Here, we study pure awareness specifically, using a psychometric approach and a philosophical background theory. We present data from the *Minimal Phenomenal Experience* (MPE) project, an interdisciplinary initiative aiming at a minimal model explanation of conscious experience, taking the phenomenal character of "pure consciousness" or "pure awareness" in meditation as its empirical entry point (for an introduction, cf. [9], sections 1 and 2).

The MPE project has two metatheoretical goals that correspond to two fundamental motivating questions: What, if anything, can count as the *simplest* form of conscious experience? And is it possible to arrive at a *minimal model explanation* for conscious experience in neurotypical human beings?

Typical examples of questions related to the first goal are: Phenomenologically, can a conscious system be exclusively aware of *awareness itself*? Please note how the concept of "awareness" we use is a phenomenological one, and not a metaphysical one (which might involve treating awareness as a Kantian "thing in itself," or even some mysteriously self-conscious

homunculus looking at awareness itself, etc.). This evidence-based phenomenological approach implies the possibility of non-egoic self-awareness, i.e., an experience of awareness itself that is not even *subjective* in the sense of being tied to a consciously experienced first-person perspective or a personal-level self-model anymore (see the discussion of Factor 3 and 8 below). If so, would there be a specific form of *phenomenal character*, a specific experiential quality that is instantiated during such episodes of pure awareness? Is there something like the *simplest* form of conscious experience, and if so, what exactly are the conceptual criteria for phenomenological "minimality"? Neuroscientifically, what are the minimally sufficient correlates for MPE to occur in neurotypical humans?

The second goal is to explore a new explanatory strategy. For empirical consciousness research, the idea is to develop a heuristic strategy of *minimalist idealization*, by describing the target phenomenon in an uncluttered way, abstracting away from everything that is not an essential feature of the core explanandum [9]. For the target of phenomenal consciousness, does consciousness *per se* have a distinct experiential character? Does it ever occur in isolation, and can it be described in a conceptually precise manner? Part of the idea is to look for the decisive explanatory and computational factors by first attempting to construct exactly solvable minimal models ([10], p. 37). Such minimal models do not aim at maximal fidelity or completeness; they are not yet mappings to fine-grained functional mechanisms, and in this sense, they are idealizations. However, they can be crucial in generating a "minimal model explanation" [10, 11]. Minimalist idealization proceeds by means of: researchers constructing and further investigating a parsimonious model of conscious experience that goes beyond mere correlation but includes only the *core causal factors* giving rise to the target phenomenon; including only those causal factors that make a difference to the *actual occurrence* and the essential phenomenal character itself; developing an idealized model of *universal and repeatable features* serving to gradually isolate the fundamental, explanatorily relevant, and structurally stable properties that underlie all different forms of conscious experience. Here, constructing such a minimal model would amount to eliminating superfluous details by extracting only the explanatorily relevant causal structure underlying the experience of awareness *per se*.

The project's basic working hypothesis is that there exists a form of "minimal phenomenal experience" (MPE hereafter; [9, 12]) that lacks time representation, spatial self-location, agency, autobiographical self-awareness, and a phenomenally experienced first-person perspective. This can be understood as an unstructured form of global content that is also devoid of perceptual, motor, affective, conceptual and propositional content.

Therefore, the project's overarching epistemic goal is to find out whether anything like "pure consciousness" really exists: Is there a state in which only consciousness *per se* is phenomenally experienced? However, as an empirical investigation, the current study does not prejudge the question of the absence of specific contents. Rather, it allows for the possibility that the non-conceptual experience of consciousness *as such* can co-occur with conscious contents. This corresponds to a theoretical treatment of MPE as a *phenomenological prototype* without sharp definitional boundaries. We also limit ourselves to one plausibly homogeneous subclass of MPE states, a paradigmatic form of "pure consciousness" experience as it appears in the context of systematic and formal contemplative practice. From the relevant body of Eastern and Western literature and based on a series of pilot studies involving committed practitioners, we extracted 92 characteristics of the phenomenology of pure consciousness experiences, which were then formed into questionnaire items. The aim of our study is to use the questionnaire data to find clusters within these items that could serve as coherent and meaningful phenomenological dimensions of the experience of pure awareness.

## Materials and methods

### Data collection

Data were collected from Jan 04, 2020 to April 01, 2020 using an online survey hosted on *SurveyGizmo* (recently renamed to *Alchemer*). In phase I, a personal invitation translated into the five survey languages and containing a link to the survey was emailed by TM to a number of regular and committed practitioners of meditation. Recipients were asked to further distribute the survey link. In phase II, the call for participation was extended, via social media, to relevant organizations and groups all over the world. According to the guidelines of the local ethics committee of the Department of Psychology of Johannes Gutenberg-Universität no formal ethics approval was necessary, because the current study satisfies all criteria for safety and clearance.

### Participants

A total of 3627 survey responses were submitted. Theoretically, the requirement for a respondent to be included in the factor analysis was to have had at least one experience of "pure awareness" (N = 2257) and to have submitted complete answers to the 92 items (N = 1171). In practice, both restrictions were slightly relaxed: 17 participants who failed to answer the screening question about pure awareness but nevertheless had complete questionnaire data were kept in the dataset. Respondents with answers to at least 86 items were also retained, as this increased the analysis sample by over 200 cases compared to those with all 92 items completed. Further lowering the minimum required number of items, however, resulted in sharply diminishing returns in terms of sample size. The total number of respondents satisfying the resulting requirements was N = 1403 (38.7%).

### Questionnaire

The questionnaire started with a screening question asking respondents whether they had ever had an experience of "pure awareness." Those answering yes were presented with the full questionnaire including some demographic and personal questions as well as the 92 questionnaire items ("MPE items"). They were instructed to select, and then focus on, only *one single, particular* experience of pure awareness that they had had and to answer all MPE items with regard to *that particular experience*. If several experiences were eligible, participants were asked to select "one in which the quality of pure awareness was particularly salient and/or one which you can remember particularly clearly". To avoid time-based response effects, presentation order of the items was randomized. At the end of the survey, participants were also asked to supply a phenomenological report about any of their experiences of pure awareness.

Respondents who had never had the requisite experience or who stated they did not understand what we meant by "pure awareness" were asked to complete only the demographic part of the survey. The demographic questions included biological sex, age, native language, nationality, country of residence, frequency and duration of meditation practice, meditation techniques, religious denomination, substance use and presence/absence of a psychiatric diagnosis.

The core of the questionnaire consisted of the 92 MPE items addressing facets of the experience of pure awareness that had previously been identified (by TM) in a comprehensive review of the literature and with the help of advanced practitioners who mostly belonged to the Vipassana and Zen traditions (similar to the final study sample, see Table 1). For example, there were items on wakefulness, mood, state of relaxation, presence of thoughts, emotions or

**Table 1. Demographic and other participant information.**

| | Analysis sample | % missing |
|---|---|---|
| N | 1403 | 0 |
| | Median (IQR, min–max) | % |
| Age | 52 (22, 17–88) | .6 |
| | N (%) | % |
| Sex | | 1.4 |
| male | 681 (48.5) | |
| female | 702 (50.0) | |
| Chosen questionnaire language | | 0 |
| German | 889 (63.4) | |
| English | 440 (31.4) | |
| Spanish | 30 (2.1) | |
| Italian | 26 (1.9) | |
| French | 18 (1.3) | |
| Native language[a] | | 1.4 |
| German | 860 (61.3) | |
| English | 215 (15.3) | |
| Dutch | 55 (3.9) | |
| Spanish | 38 (2.7) | |
| Italian | 36 (2.6) | |
| French | 22 (1.6) | |
| Russian | 13 (.9) | |
| Swedish | 13 (.9) | |
| Polish | 11 (.8) | |
| Hindi | 10 (.7) | |
| Portuguese | 10 (.7) | |
| Other | 100 (7.1) | |
| Country of residence | | 1.1 |
| Germany | 777 (55.4) | |
| USA | 110 (7.8) | |
| Switzerland | 87 (6.2) | |
| Netherlands | 63 (4.5) | |
| United Kingdom | 63 (4.5) | |
| Italy | 28 (2.0) | |
| Austria | 25 (1.8) | |
| Australia | 19 (1.4) | |
| France | 18 (1.3) | |
| Canada | 16 (1.1) | |
| Spain | 15 (1.1) | |
| India | 13 (0.9) | |
| Sweden | 12 (0.9) | |
| Belgium | 10 (0.7) | |
| Denmark | 10 (0.7) | |
| Nepal | 10 (0.7) | |
| Other | 112 (8.0) | |
| Nationality | | 1.2 |
| Germany | 786 (56.0) | |
| USA | 103 (7.3) | |

(*Continued*)

**Table 1.** (Continued)

| | Analysis sample | % missing |
|---|---|---|
| Switzerland | 84 (6.0) | |
| United Kingdom | 59 (4.2) | |
| Netherlands | 57 (4.1) | |
| Italy | 38 (2.7) | |
| Austria | 24 (1.7) | |
| Australia | 17 (1.2) | |
| India | 16 (1.1) | |
| France | 15 (1.1) | |
| Sweden | 15 (1.1) | |
| Canada | 14 (1.0) | |
| Spain | 13 (0.9) | |
| Other | 145 (10.3) | |
| *Meditation experience* | | |
| Regular meditators | 1085 (77.3) | 1.3 |
| | Median (IQR, min–max) | |
| Years meditating | 10 (21, 0–55) | 5.2 |
| Sessions per day | 1 (1, 0–60) | 6.8 |
| Minutes per session | 30 (20, 1–240) | 9.9 |
| Meditation technique[a] | | 2.7 |
| Vipassana | 616 (43.9) | |
| Other | 603 (43.0) | |
| Zen | 490 (34.9) | |
| Metta | 399 (28.4) | |
| Mahamudra/Dzogchen | 275 (19.6) | |
| MBSR | 274 (19.5) | |
| Shamata | 272 (19.4) | |
| TM | 192 (13.7) | |
| N of meditation techniques practiced | | 2.7 |
| 1 | 558 (39.8) | |
| 2 | 280 (20.0) | |
| 3 | 253 (18.0) | |
| 4 | 176 (12.5) | |
| 5 | 67 (4.8) | |
| 6 | 17 (1.2) | |
| 7 | 9 (.6) | |
| 8 | 5 (.4) | |
| Substance use[a] | | 2.3 |
| None | 1161 (82.8) | |
| Cannabis | 123 (8.8) | |
| Psychedelics | 96 (6.9) | |
| Other | 46 (3.3) | |
| Entactogens / MDMA | 37 (2.6) | |
| Stimulants | 25 (1.8) | |
| Anesthetics or Dissociatives | 13 (.9) | |
| Sedatives / Tranquilizers | 13 (.9) | |
| Opiates | 4 (.3) | |
| GHB / GBL | 2 (.1) | |

(*Continued*)

**Table 1.** (Continued)

| | Analysis sample | % missing |
|---|---|---|
| Psychiatric diagnosis | | 1.1 |
| Yes | 90 (6.4) | |
| No | 1297 (92.4) | |
| Religious denomination[a] | | 1.6 |
| "Spiritual but not religious" (SBNR) / "Spiritual but not affiliated" (SBNA) | 639 (45.6) | |
| Buddhist | 322 (23.0) | |
| Christian | 320 (22.8) | |
| Secular | 188 (13.4) | |
| Other | 72 (5.1) | |
| Hindu | 33 (2.4) | |
| Jewish | 7 (.5) | |
| Muslim | 4 (.3) | |
| | Median (IQR, min–max) | |
| Importance of meditation (0–100) | 90 (23, 0–100) | 4.1 |
| Importance of religion (0–100) | 50 (78, 0–100) | 10.1 |

[a]Multiple responses were possible. Percentages do not sum to 100%.

IQR = Interquartile Range, max = maximum, MBSR = Mindfulness-Based Stress Reduction, min = minimum, N = Number, TM = Transcendental Meditation.

sensory perceptions, presence of a sense of self and the experience of time, to name just a few. The full list of items can be found in Table 6.

MPE items were answered on a visual analogue scale ranging from 0 to 100, indicating the degree to which a particular facet of experience was present. In practice, respondents used their computer mouse to position a slider on a horizontal line (the slider defaulted to the mid-point of the line). Scale endpoints were given labels appropriate to the form of each question. Not all items therefore had the same response labels. In questions about the presence and intensity of a particular phenomenal state (e.g. "Was there a sense of self?" or "Did you have a visual experience of brightness with closed eyes?"), scale anchors were "No" on the left end and "Very much so" or "Very strongly" on the right end. In questions about the presence and frequency of certain countable experiential elements (e.g. "Did you have thoughts?" or "Did you have memories?") the label changed to "Very many" on the right end. Occasionally, there were still other response labels when an item required it (e.g. the question "How alert were you" had the anchors 0 = "unconscious", 50 = "normal daytime alertness" and 100 = "much more alert than normal").

Although items were presented in a form answerable on a continuous scale, not all items lent themselves to this approach equally well. For some, dichotomous "yes" / "no" anchors seemed most appropriate, although even in these cases, a continuous degree of affirmability was assumed to exist between the binary end points. Example questions include "Did you know that you would be able to deliberately think thoughts if you wanted to?" and "Would it be a good description to say that your experience of pure awareness, upon self-recognizing, also recognized itself as that awareness which had always been present in all experiences in the past?".

To maximize the number of complete answers, items were "soft-required," i.e. when skipped deliberately or accidentally, a pop-up message reminded respondents to provide an answer if possible. A strict response enforcement was considered counterproductive, as respondents might legitimately find it hard or even impossible to answer certain questions.

The first 1000 respondents to completely answer all questions were offered a monetary reward of € 50. The money was transferred using the web service *Transferwise*. Interested participants were asked to leave a valid email address, as this was all the service required.

The questionnaire contained one control item (#66), which was a near-duplicate of another item (#42) about the non-visual experience of "radiance". It was included to provide a sense of the reliability of item responses.

## Translations

Five different language versions of the MPE-92M exist. The original version was developed in English and subsequently translated into German, French, Italian and Spanish. To create the translations, a bilingual speaker translated from the English original into a second language, and then translated back into English. The original and back-translated English versions were then compared, and adjustments were made if necessary. In some cases, this process required several iterations.

## Statistics

Bartlett's test for sphericity ($p < 0.001$) and the Kaiser-Meyer-Olkin Measure of Sampling Adequacy (.94) confirmed the basic suitability of the data for factor analysis. Due to non-normality of item distributions, exploratory factor analysis (EFA) was performed on Spearman correlations using principal-factor estimation. Missing values were deleted pairwise, and the average number of participants available across all pairwise correlations (N = 1393) was specified for factor analysis. Loadings were rotated obliquely by the *quartimin* method in order to allow for correlated factors. Theoretical and conceptual considerations as well as several statistical methods were used to guide the number of factors to be extracted. These methods included the scree plot, the Kaiser criterion (eigenvalue > 1), Horn's parallel analysis, Velicer's MAP criterion and the Bayesian Information Criterion (BIC). Only items with primary loadings of 0.3 or higher were assigned to their corresponding factors.

To explore the fit of the EFA solution, exploratory factor analysis in the context of confirmatory factor analysis (E/CFA) was performed. The strongest-loading items per factor were defined as anchor items whose cross-loadings were fixed at zero, while all other items were allowed to freely load on all factors. A further identification constraint was to fix factor variances at unity. E/CFA produces a saturated solution for the loading structure, so that modification indexes are available only for error covariances. Correlated errors were tentatively allowed when indicated by modification indexes corresponding to standardized expected parameter changes of $\geq 0.2$. Confirmatory factor analysis (CFA) was performed only to assess the fit of any potential E/CFA solution, on which simple structure was imposed by constraining each item to load exclusively on its primary factor. Again, factor variances were set to 1 to allow identification. Internal consistencies per factor were calculated using Cronbach's alpha.

We refrain from presenting an "optimized" short-form version of the questionnaire. Although popular, attempts to construct and evaluate a reduced item pool in a dataset resulting from the larger original item pool are in general methodologically unsound and have been strongly advised against [13–15].

Fit indexes were interpreted according to recommendations in Brown [16]. Absolute fit was assessed by the standardized root mean square residual (SRMR) and the root mean square error of approximation (RMSEA). The SRMR indicates the average discrepancy between observed and model-predicted item correlations. The RMSEA indicates deviations of the fitting function from perfect fit and incorporates a penalty for lack of model parsimony. Relative fit was assessed by the comparative fit index (CFI) and the Tucker-Lewis index (TLI). The CFI

compares a given model to a baseline model that specifies no correlations between items. The TLI is similar to the CFI, but again includes a penalty for lack of model parsimony. The following values were considered to be consistent with good fit: SRMR ≤ .08, RMSEA ≤ .05, CFI and TLI ≥ .95. To account for non-normality in the data, computations of model fit were repeated using quasi maximum likelihood (QML) estimation and the Satorra-Bentler correction [17]. Analyses were performed in Stata 14.2 for Mac [18].

## Results

### Demographics of analysis sample (N = 1403)

Participants' mean age was 52 years. The sample was nearly equally split between men and women (48.5% vs 50.0%). The questionnaire language chosen most frequently was German (63.4%), followed by English (31.4%). Correspondingly, German was the most frequently reported native language (61.3%), again followed by English (15.3%). A total of 57 nations were represented among participants. Regular meditators constituted the majority (77.3%) of the sample. The most frequently named religious affiliation was "Spiritual but not religious" / "Spiritual but not affiliated." Substance use and psychiatric diagnoses were nearly absent. Complete demographic information is shown in Table 1.

### Distribution of meditation techniques

Overall, Vipassana was the most frequently named meditation technique (over 600 reports). As shown in Fig 1, the single most frequent constellation of techniques was "other" (235 participants), exclusive Zen meditation (138 participants), and exclusive TM (97 participants).

### Comparing respondent groups: Those with vs. those without an experience of pure awareness

A total of 2466 participants answered the screening question. The vast majority (91.5%, N = 2257) reported having had at least one experience of pure awareness. In multivariable regression, speaking German or Spanish as well as every additional ten years of meditation practice increased the likelihood of a positive response by 2–5%. Being male reduced the likelihood of a positive response by 4% (Table 2).

### Comparing respondent groups: Those with an experience of pure awareness vs. those not understanding the concept of pure awareness

3.2% (N = 78) of participants reported not understanding the concept of pure awareness. In multivariable analysis, speaking German and practicing Mahamudra/Dzogchen were the only statistically significant predictors of understanding and experiencing pure awareness compared to not understanding it. Again, effects were small, raising the likelihood by 3% (Table 3).

### Comparing respondent groups: Those included vs those not included in factor analysis

Inclusion in the final analysis sample required at least one experience of pure awareness and a nearly complete set of MPE items (at least 86 out of 92 items). 1403 participants met these requirements, while 2264 did not. In multivariable analysis, regular meditation, every additional ten years of meditation practice, speaking German and practicing Shamata, Zen or some "other", unlisted meditation technique increased the likelihood of being included in factor analysis by 3–9%. Being male reduced the likelihood by 7% (Table 4).

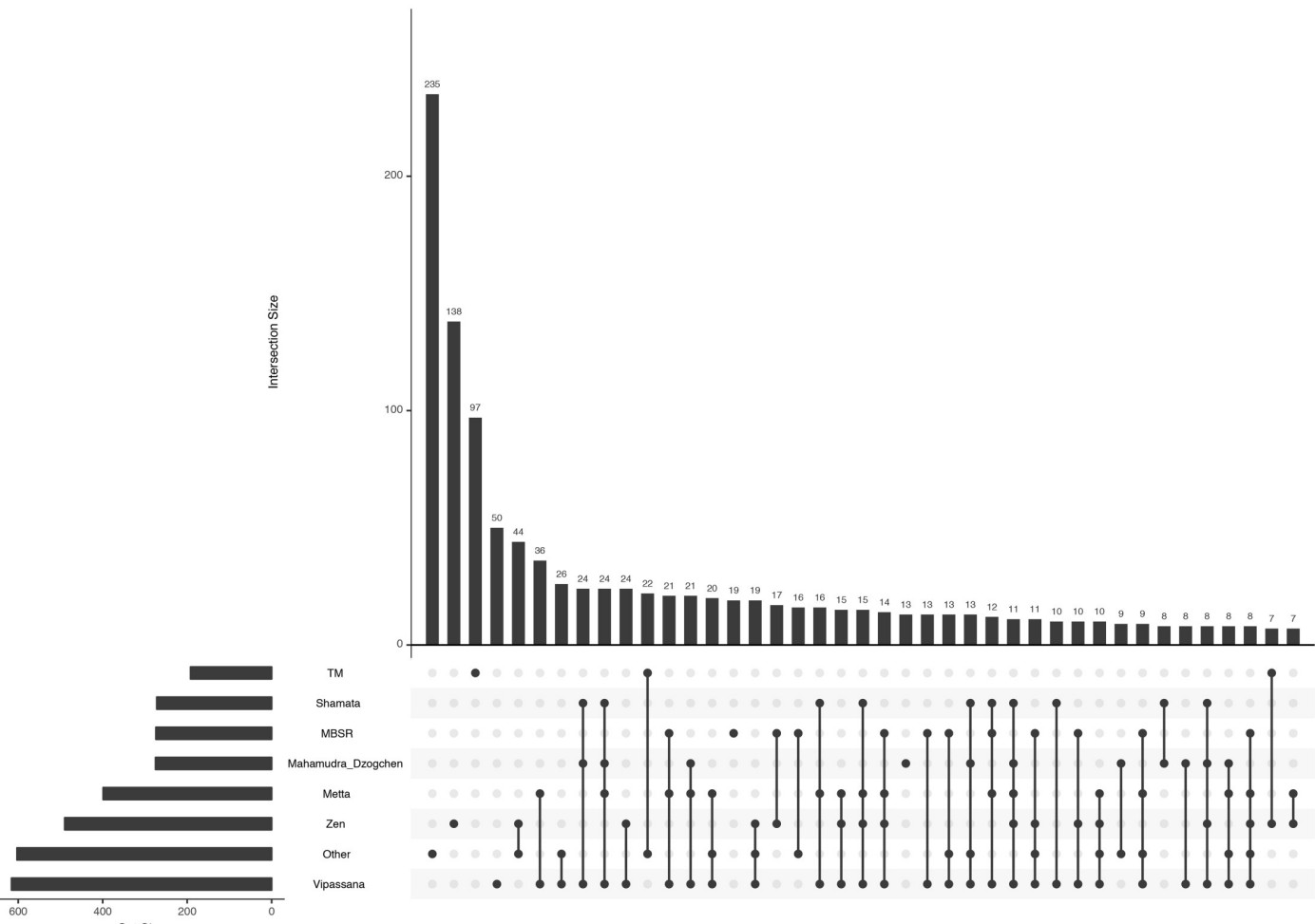

**Fig 1. Combinations of meditation techniques.** Upset plot showing the combinations of meditation techniques occurring in the analysis sample (N = 1403). The top graph is a histogram, with each bar representing the frequency of a particular (constellation of) meditation technique(s) that is/are identified right below it. Black dots connected by black lines string together meditation techniques that belong to a given constellation. The bar chart on the left shows the overall frequencies of each meditation technique.

### Reliability: Control item

In about half of the 1403 participants, scores on the two almost identical items 42 and 66 diverged by 10 or less (Fig 2), with possible differences ranging from 0 to 100.

### MPE items: Descriptive analysis

MPE items were generally non-normally distributed. Responses tended to cluster at the extreme ends of the scale (Fig 3). The highest average ratings were given to items related to wellbeing and relaxation; the overall lowest rating concerned the presence of pain during the experience.

### Exploratory factor analysis

The number of participants available for factor analysis was 1393. This number is an average over the number of participants available for each pairwise item correlation, which ranged

**Table 2. Risk ratios comparing participants with (N = 2257) and without (N = 131) an experience of pure awareness.**

| | Unadjusted Risk Ratio [95% C.I.][a] | Adjusted Risk Ratio [95% C.I.][b] | % missing | |
|---|---|---|---|---|
| | | | pure awareness (N = 2257) | no pure awareness (N = 131) |
| Age (10-year units) | 1.02 [1.01–1.02]*** | 1.00 [0.99–1.01] | 16.0 | 11.5 |
| Male sex | 0.94 [0.92–0.96]*** | 0.96 [0.94–0.98]*** | 16.2 | 12.2 |
| Meditates regularly | 1.03 [1.00–1.06]* | 1.02 [0.99–1.04] | 19.6 | 19.1 |
| Years of practice (10-year units) | 1.02 [1.01–1.03]*** | 1.02 [1.01–1.03]*** | 23.0 | 31.3 |
| Questionnaire language | | | 0 | 0 |
| English | 0.96 [0.93–0.98]*** | N.A.[c] | | |
| French | 0.94 [0.83–1.07] | 0.96 [0.84–1.11] | | |
| German | 1.04 [1.02–1.06]*** | 1.04 [1.01–1.07]** | | |
| Italian | 1.00 [0.94–1.07] | 1.01 [0.93–1.10] | | |
| Spanish | 1.03 [0.98–1.07] | 1.05 [1.00–1.10]* | | |
| Meditation technique | | | 21.9 | 29.0 |
| Mahamudra/Dzogchen | 1.01 [0.99–1.04] | 1.00 [0.97–1.03] | | |
| MBSR | 1.03 [1.01–1.05]* | 1.02 [1.00–1.04] | | |
| Metta | 1.01 [0.98–1.03] | 1.00 [0.97–1.02] | | |
| Shamata | 1.03 [1.00–1.05]* | 1.02 [0.99–1.05] | | |
| TM | 1.03 [1.00–1.05]* | 1.00 [0.97–1.03] | | |
| Vipassana | 1.01 [0.99–1.03] | 1.01 [0.98–1.03] | | |
| Zen | 1.01 [0.99–1.03] | 1.00 [0.98–1.02] | | |
| Other | 1.03 [1.01–1.05]** | 1.02 [1.00–1.04] | | |

[a]Univariable general linear regressions with Gaussian distribution, log link and robust standard errors, variable N (1827–2388).

[b]Multivariable general linear regression with Gaussian distribution, log link and robust standard errors, N = 1750.

[c]English is the reference category.

*p < .05,

**p < .01,

***p < .001.

N.A. = not applicable.

from 1357 to 1403. Fig 4 shows factor solutions with 6 to 12 factors obtained by oblique rotation.

Different statistical criteria for the optimal number of factors yielded different recommendations. While a scree plot suggested a 4- or 5-factor solution, Velicer's MAP criterion and the Kaiser "eigenvalue > 1" rule suggested 9 factors, Horn's parallel analysis suggested 13 factors and the Bayesian Information Criterion (BIC) was optimal at 14 factors. The latter, however, is computed by maximum likelihood estimation under the assumption of multivariate normality, which does not obtain in our data (see Fig 3).

From a conceptual and theoretical point of view, a solution with fewer than 6 factors was deemed not sufficiently fine-grained to discriminate between what we considered to be clearly distinct phenomenological aspects of MPE. A 12-factor solution, based on a loading threshold of 0.3, was finally chosen as providing an optimal balance between conceptual resolution and robustness of factors in terms of the number of supporting items.

The factors were named "Time, Effort and Desire," "Peace, Bliss and Silence," "Self-Knowledge, Autonomous Cognizance and Insight," "Wakeful Presence," "Pure Awareness in Dream and Sleep," "Luminosity," "Thoughts and Feelings," "Emptiness and Non-egoic Self-awareness," "Sensory Perception in Body and Space," "Touching World and Self," "Mental Agency" and "Witness Consciousness." Internal consistencies ranged between 0.52 and 0.82, with a

**Table 3. Risk ratios comparing participants with an experience of pure awareness (N = 2257) to those not understanding the concept of pure awareness (N = 78).**

| | Unadjusted Risk Ratio [95% C. I.][a] | Adjusted Risk Ratio [95% C. I.][b] | % missing | |
|---|---|---|---|---|
| | | | pure awareness (N = 2257) | did not understand concept (N = 78) |
| Age (10-year units) | 1.00 [1.00–1.01] | 0.99 [0.98–1.00] | 15.6 | 20.5 |
| Male sex | 0.99 [0.97-.1.01] | 0.99 [0.98–1.01] | 16.2 | 19.3 |
| Meditates regularly | 1.02 [1.00–1.05]* | 1.02 [0.99–1.04] | 19.6 | 21.8 |
| Years of practice (10-year units) | 1.00 [1.00–1.01] | 1.01 [1.00–1.01] | 23.0 | 30.8 |
| Questionnaire language | | | 0 | 0 |
| English | 0.98 [0.96–0.99]** | N.A.[c] | | |
| French | 0.99 [0.92–1.07] | 0.99 [0.89–1.11] | | |
| German | 1.03 [1.01–1.04]** | 1.03 [1.01–1.06]*** | | |
| Italian | 1.00 [0.95–1.05] | 1.01 [0.93–1.08] | | |
| Spanish | 0.97 [0.92–1.03] | 1.03 [0.98–1.07] | | |
| Meditation technique | | | 21.9 | 35.9 |
| Mahamudra/Dzogchen | 1.01 [1.00–1.0] | 1.03 [1.00–1.05]* | | |
| MBSR | 1.00 [0.98–1.02] | 1.00 [0.98–1.02] | | |
| Metta | 1.00 [0.99–1.02] | 1.01 [0.99–1.03] | | |
| Shamata | 0.99 [0.97–1.02] | 0.99 [0.96–1.01] | | |
| TM | 1.01 [0.99–1.03] | 1.00 [0.98–1.03] | | |
| Vipassana | 0.99 [0.98–1.01] | 0.99 [0.97–1.01] | | |
| Zen | 1.00 [0.99–1.02] | 1.00 [0.99–1.02] | | |
| Other | 1.01 [1.00–1.03] | 1.01 [1.00–1.03] | | |

[a]Univariable general linear regressions with Gaussian distribution, log link, and robust standard errors, variable N (1791–2335).

[b]Multivariable general linear regression with Gaussian distribution, log link, and robust standard errors, N = 1717.

[c]English is the reference category.

*p < .05,

**p < .01,

***p < .001.

N.A. = not applicable.

mean of 0.72. Nine out of 12 factors had consistencies of $\geq 0.70$. Table 5 shows factor names, the proportion of explained variance per factor, and internal consistencies.

Table 6 shows item-to-factor assignments, factor loadings, factor names and item uniqueness of the 12-factor solution. Uniqueness values, which represent unique, non-shared item variance, were relatively high, with 32 (35%) of 92 items above 0.6. Correspondingly, total variance explained by common factors was at a moderate 44%.

Factor correlations, which are a consequence of oblique rotation, are shown in Table 7.

## Confirmatory factor analysis

Overall, model fit of the solution from exploratory factor analysis (EFA) was moderate to acceptable, but significantly degraded after imposing simple structure in a confirmatory factor analysis (CFA).

A reproduction of the 12-factor solution using EFA in the context of CFA produced a moderate fit to the data: while absolute fit indexes were consistent with good fit (RMSEA = .034, 90% CI = [.033–.035]; p(RMSEA $\leq$ .05) = 1.0; SRMR = .025), comparative fit was weak (CFI = .875; TLI = .834). Using the Satorra-Bentler correction slightly improved fit (RMSEA = .030;

**Table 4. Risk ratios comparing participants included (N = 1403) to those not included (N = 2224) in factor analysis.**

| | Unadjusted Risk Ratio [95% C.I.][a] | Adjusted Risk Ratio [95% C.I.][b] | % missing | |
|---|---|---|---|---|
| | | | included in FA (N = 1403) | excluded from FA (N = 2224) |
| Age (10-year units) | 1.04 [1.02–1.06]*** | 0.99 [0.96–1.01] | 0.6 | 68.4 |
| Male sex | 0.94 [0.88–1.00]* | 0.93 [0.87–0.99]* | 1.4 | 68.2 |
| Meditates regularly | 1.13 [1.05–1.22]*** | 1.08 [1.00–1.17]* | 1.3 | 72.2 |
| Years of practice (10 year units) | 1.04 [1.02–1.06]*** | 1.03 [1.00–1.06]* | 5.2 | 74.4 |
| Questionnaire language | | | 0 | 0 |
| English | 0.88 [0.80–0.96]** | N.A.[c] | | |
| French | 0.86 [0.59–1.26] | 1.20 [0.98–1.46] | | |
| German | 1.17 [1.08–1.28]*** | 1.09 [1.02–1.17]* | | |
| Italian | 1.00 [0.74–1.36] | 0.88 [0.68–1.15] | | |
| Spanish | 0.76 [0.56–1.03] | 0.90 [0.71–1.14] | | |
| Meditation technique | | | 2.7 | 74.7 |
| Mahamudra/Dzogchen | 1.05 [0.98–1.12] | 1.03 [0.96–1.11] | | |
| MBSR | 1.03 [0.96–1.10] | 1.02 [0.95–1.10] | | |
| Metta | 1.03 [0.96–1.09] | 1.01 [0.94–1.18] | | |
| Shamata | 1.09 [1.02–1.27]* | 1.09 [1.01–1.18]* | | |
| TM | 1.07 [0.99–1.15] | 1.06 [0.97–1.15] | | |
| Vipassana | 1.01 [0.96–1.07] | 1.02 [0.96–1.10] | | |
| Zen | 1.06 [1.00–1.12]* | 1.08 [1.02–1.14]* | | |
| Other | 1.06 [1.00–1.12] | 1.08 [1.02–1.15]** | | |

[a]Univariable general linear regressions with Gaussian distribution, log link and robust standard errors, variable N (1900–3627).

[b]Multivariable general linear regression with Gaussian distribution, log link and robust standard errors, N = 1814.

[c]English is the reference category.

*p < .05,

**p < .01,

***p < .001.

FA = factor analysis, N = number, N.A. = not applicable.

SRMR = .025; CFI = .887; TLI = .849). Model fit improved more noticeably after allowing correlated errors as indicated by modification indexes corresponding to expected parameter changes of ≥ .2: RMSEA = .025, 90% CI = [.023–.026]; p(RMSEA ≤ .05) = 1.0; SRMR = .022; CFI = .935; TLI = .913. Applying the Satorra-Bentler correction further improved fit very slightly: RMSEA = .021; SRMR = .022; CFI = .946; TLI = .928.

After imposing simple structure by suppressing cross-loadings, model fit decreased substantially. While absolute fit was still acceptable (RMSEA = .053, 90% CI = [.052–.054]; p (RMSEA ≤ .05) = 0.00; SRMR = .075), comparative fit was very weak (CFI = .707; TLI = .690). Comparative fit improved marginally after applying the Satorra-Bentler correction (CFI = .715; TLI = .699). Allowing correlated errors as indicated by modification indexes corresponding to expected parameter changes of ≥ .2 improved the model, but comparative fit remained weak (RMSEA = .004, 90% CI = [.039–.041]; p(RMSEA ≤ .05) = 1.0; SRMR = .060; CFI = .840; TLI = .826). Marginal improvement resulted from applying the Satorra-Bentler correction (RMSEA = .036; SRMR = .060; CFI = .849; TLI = .836).

## Phenomenological reports

A total of 1171 reports about experiences of pure awareness were submitted, of which 841 were usable. These will undergo a qualitative analysis, to be published separately.

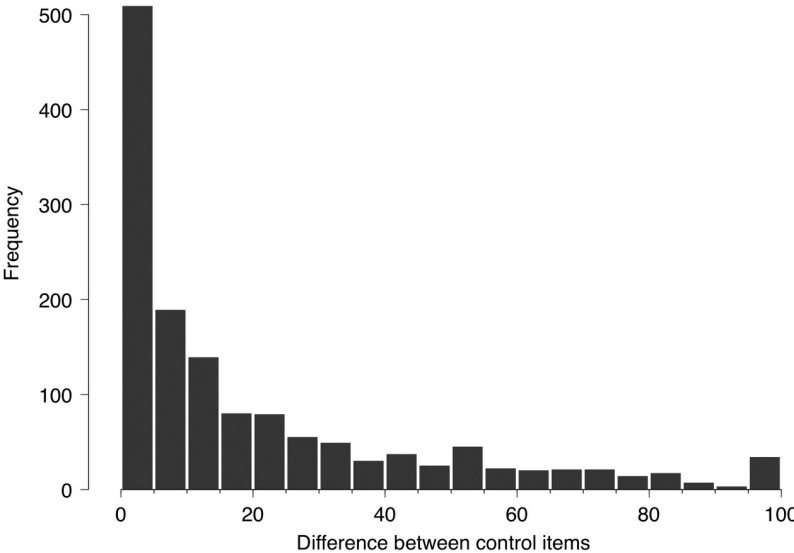

**Fig 2. Distribution of difference between control items.** Distribution of differences in item scores between two almost identically worded questions (#42 and #66) asking about the presence of a non-visual experience of "radiance". Item scores range from 0–100, so that the maximum possible difference is 100. N = 1403.

## Discussion

The experience of "pure awareness" has been reported and described in the literature on contemplative practices since at least around the 8th to 6th century BCE ([19], p39). In this study, we attempted a first phenomenological characterization of such experiences using a 92-item online questionnaire distributed among meditators. Over 1400 responses were gathered and factor-analyzed. The preferred solution contained 12 factors accounting for a moderate 44% of the total variance, with unique item variances being relatively large. Therefore, the chosen solution is likely to leave some of the questionnaire's phenomenological and semantic content as well as survey response patterns unexplained. In line with this, reproducing the 12-factor solution in the context of a confirmatory factor analysis resulted in a moderate-to-acceptable fit to the data, while imposing simple structure produced an ill-fitting solution. It is clear that the chosen factor solution will not fit the present data well without allowing for a significant amount of cross-loading.

With these limitations in mind, we now give possible interpretations of the 12 factors in terms of their phenomenology and our background theory. We also make connections to how this phenomenology may typically manifest in meditative practice. The interpretations and connections we propose are predicated on the assumption that these factors or very similar ones may be reproduced in future studies, which is by no means guaranteed. However, such speculation may be fruitful for future research by giving rise to new testable hypotheses concerning the phenomenal structure of pure awareness experiences.

**Factor 1** ("Time, Effort and Desire") bundles experiential aspects of a process that could be described as "dual mindfulness." Phenomenologically, there is still a meditator and a goal state, there is a sense of effort created by either mental or bodily agency, and accordingly the subjective experience of time emerges. **Factor 1** points to the conscious experience of a process that is still goal-directed (there is a desire), and in which there is a sense of effort (for example, in controlling body posture, or in the subjective experience of repeatedly bringing attention back to the breath or other objects of meditation, as in Shikantaza, in Shamatha practice or in

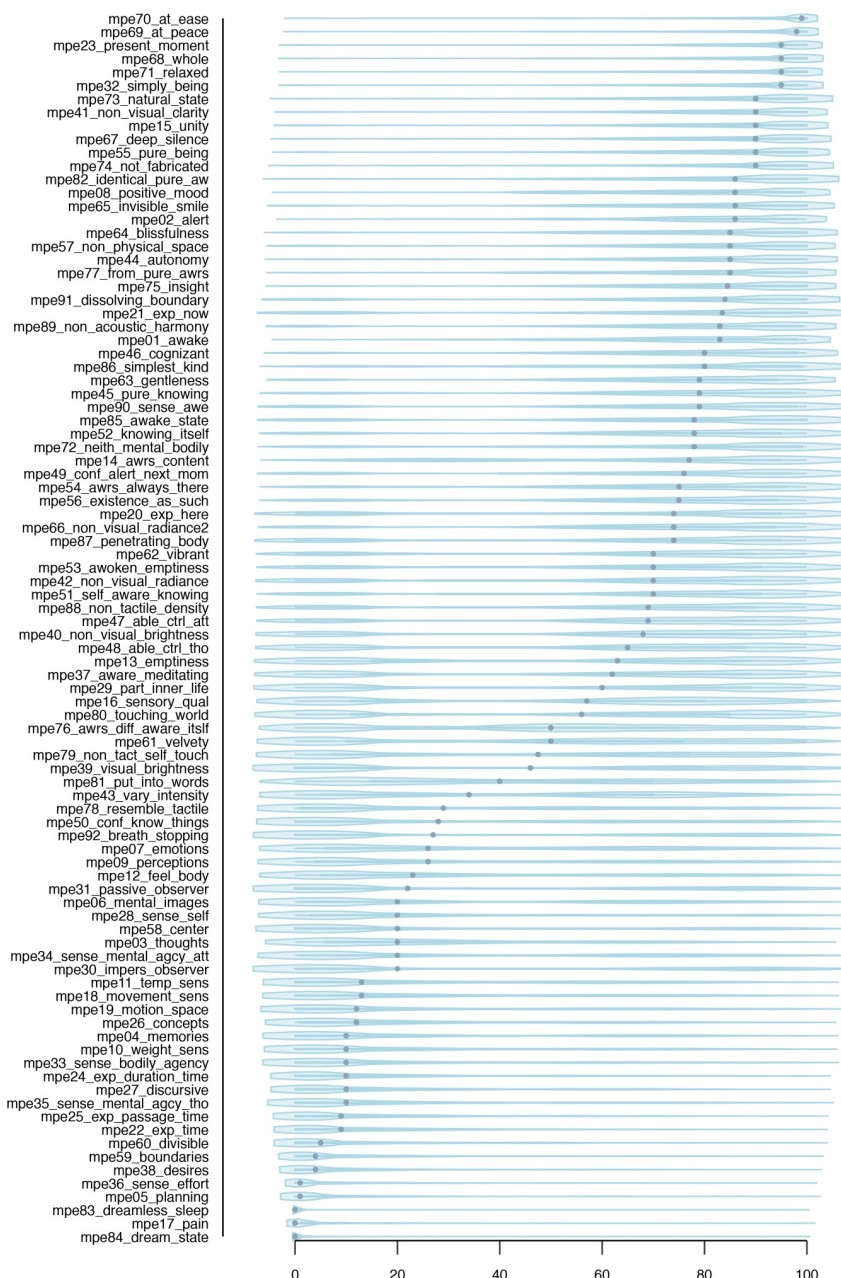

**Fig 3. Distribution of questionnaire items.** Statistical distribution of questionnaire items in analysis sample (N = 1403). Dots indicate medians.

other forms of focussed-attention meditation). Typically, there will be an attentional lapse, followed by the phenomenology of noticing, remembering the goal state and re-focussing [20, 21]. As a result, temporal experience is preserved: for example, we find the phenomenology of duration and of time passing. There may already be an experience of awareness as such, but this is still a *meditation* experience, not yet effortless and not a full-absorption episode. (A "full-absorption episode" is defined as an experience in which MPE is the only kind of phenomenal character that can later be reported, i.e., a phenomenal state in which the experiential

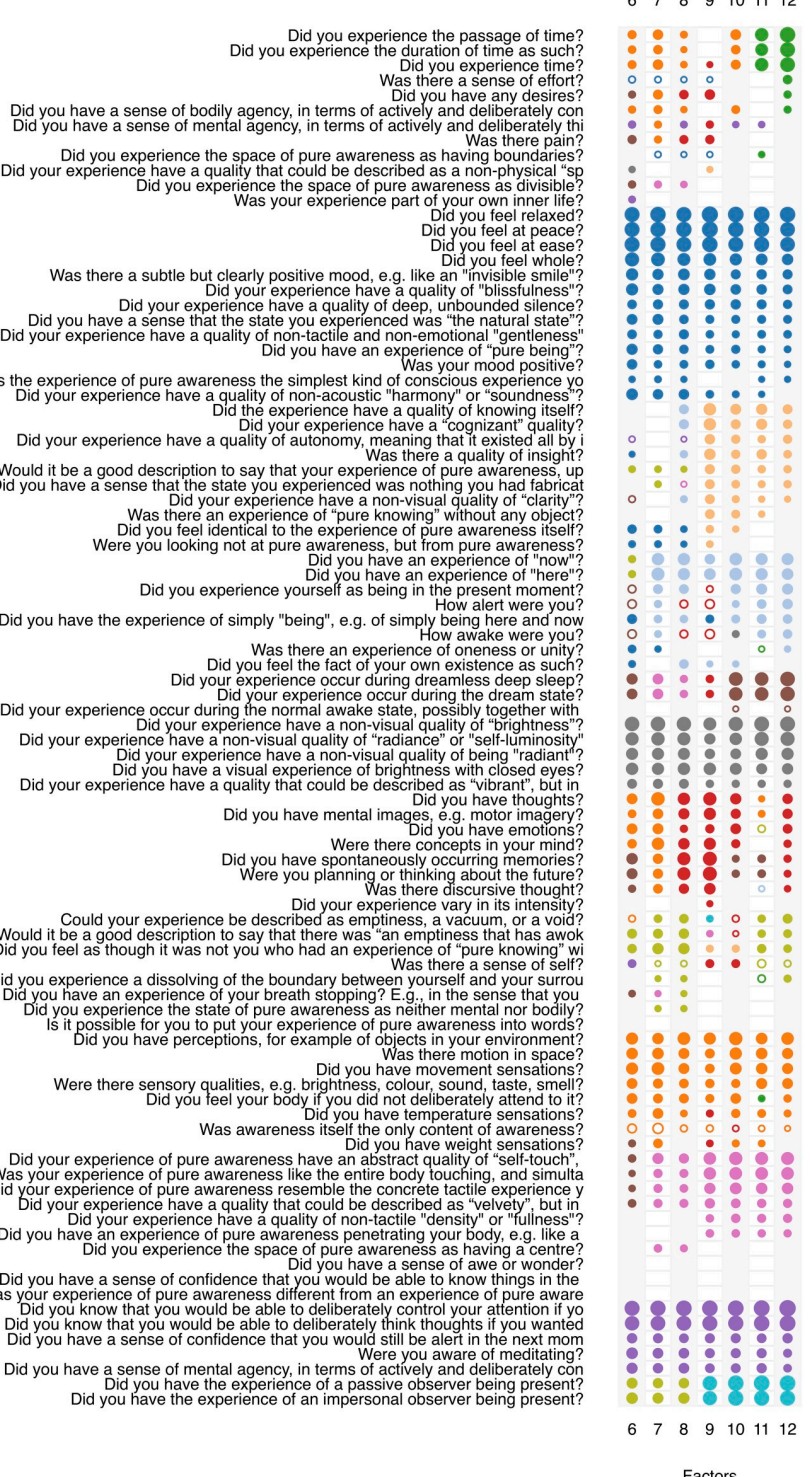

**Fig 4. Results of exploratory factor analysis of the MPE-92M questionnaire (N = 1393).** Questionnaire items are listed on the y-axis. Each column corresponds to a full factor solution with the number of extracted factors given on the x-axis. The size (area) of the circles is proportional to the factor loadings. Open circles indicate negative loadings, and loadings below 0.3 are not shown. Color represents factor membership. An attempt was made to identify the "same" factors across all factor solutions and color them identically. "Sameness" here is not objectively definable, but the degree of similarity was operationalized as membership in the same cluster following hierarchical cluster analysis. Items are sorted by factor membership and size of factor loading in the 12-factor solution, as it was this solution that was finally chosen as optimal.

**Table 5. Factor names and explained variance per factor.**

| Factor number | Factor name | % explained variance[a] | Internal consistency[b] |
|---|---|---|---|
| 1 | Time, Effort and Desire | 27 | 0.79 |
| 2 | Peace, Bliss and Silence | 25 | 0.82 |
| 3 | Self-Knowledge, Autonomous Cognizance and Insight | 16 | 0.70 |
| 4 | Wakeful Presence | 15 | 0.67 |
| 5 | Pure Awareness in Dream and Sleep | 15 | 0.52 |
| 6 | Luminosity | 14 | 0.76 |
| 7 | Thoughts and Feelings | 13 | 0.81 |
| 8 | Emptiness and Non-egoic Self-awareness | 13 | 0.65 |
| 9 | Sensory Perception in Body and Space | 12 | 0.71 |
| 10 | Touching World and Self | 12 | 0.72 |
| 11 | Mental Agency | 9 | 0.74 |
| 12 | Witness Consciousness | 7 | 0.82 |

[a]Factors after oblique rotation are correlated and therefore account for overlapping portions of total item variance. Proportions of explained variance therefore cannot be summed up across factors. Instead, the total explained variance of the entire factor solution can be obtained as average item communality, which was 44%.
[b]Cronbach's alpha.

quality of "pure consciousness" is the only one that is later available for memory and verbal report).

Factor correlations show that "Time, Effort and Desire" is negatively correlated with "Peace, Bliss and Silence" (**Factor 2**) and "Emptiness and Non-egoic Self-awareness" (**Factor 8**). This makes good phenomenological sense, because these factors express deep relaxation, existential ease, a sense of wholeness, positive affect and the atemporal, non-dual and selfless quality of "pure knowing" (see below). "Time, Effort and Desire" is positively correlated with the arising of thoughts and feelings, with mind-wandering, discursive thought and different forms of mental time travel (**Factor 7**) and also with the noticing of perceptual content arising (**Factor 9**). Our data and a qualitative assessment of verbal reports (to be published in a separate monograph) show how, in a large sample, even experienced meditators and committed practitioners have varied experiences, including frequent attentional lapses and other banal or commonplace states. This point also relates to the existence of very short spontaneous occurrences of MPE in everyday life, and to the micro-meditative technique of "glimpsing" outside of formal practice, in both of which states of pure consciousness are sandwiched by longer periods of more ordinary phenomenology. All of this is part of a normal meditation experience, and to be expected.

What we have defined as "dual mindfulness" (i.e., a process in which there is still a meditator, a goal state, a sense of effort, and the subjective experience of time) is directly related to a specific semantic constraint for the concept of MPE which was extracted from the existing literature in an earlier study, namely "Introspective Availability" (PC4; cf. [9]; section 2.2.1). One traditional assumption, extensively discussed in Eastern as well as in Western philosophy of mind, is that we can sometimes actively direct introspective attention to the quality of consciousness *per se*. If this turns out to be correct, then one can distinguish different states by the degree of actually ongoing introspective access to precisely this quality.

**Factor 2** ("Peace, Bliss and Silence") picks out the experience of relaxation and ease, which is perhaps the best-known effect of meditation practice generally. This factor also refers to a simple experience of deep, unbounded silence (cf. [22]) and "pure being", which is described as natural and gentle. The two items relating to the phenomenal experience of "peace" and

**Table 6. Factor membership and loadings of MPE items in 12-factor solution (with color coding corresponding to Fig 4).**

| Item Nr | Item long name | Factor 1 Time, Effort and Desire | Factor 2 Peace, Bliss and Silence | Factor 3 Self-Knowledge, Autonomous Cognizance and Insight | Factor 4 Wakeful Presence | Factor 5 Pure Awareness in Dream and Sleep | Factor 6 Luminosity | Factor 7 Thoughts and Feelings | Factor 8 Emptiness and Non-egoic Self-awareness | Factor 9 Sensory Perception in Body and Space | Factor 10 Touching World and Self | Factor 11 Mental Agency | Factor 12 Witness Consciousness | Uniqueness |
|---|---|---|---|---|---|---|---|---|---|---|---|---|---|---|
| 25 | Did you experience the passage of time? | 0.76 | -0.05 | 0.00 | 0.00 | -0.02 | 0.02 | -0.01 | 0.01 | 0.09 | -0.02 | 0.01 | 0.03 | 0.34 |
| 24 | Did you experience the duration of time as such? | 0.73 | -0.01 | 0.00 | 0.04 | 0.00 | 0.01 | 0.01 | 0.00 | 0.04 | 0.00 | -0.06 | 0.04 | 0.45 |
| 22 | Did you experience time? | 0.73 | -0.02 | -0.01 | -0.04 | -0.02 | 0.00 | 0.00 | -0.07 | 0.08 | -0.03 | 0.01 | 0.01 | 0.35 |
| 36 | Was there a sense of effort? | 0.44 | -0.21 | -0.14 | -0.06 | 0.10 | 0.03 | 0.14 | 0.08 | -0.02 | 0.02 | 0.05 | 0.11 | 0.43 |
| 38 | Did you have any desires? | 0.37 | -0.14 | -0.07 | -0.10 | 0.12 | 0.01 | 0.28 | -0.01 | 0.07 | 0.03 | 0.03 | 0.08 | 0.40 |
| 33 | Did you have a sense of bodily agency, in terms of actively and deliberately controlling your body movements? | 0.36 | -0.11 | -0.10 | 0.08 | 0.03 | 0.00 | 0.09 | 0.02 | 0.21 | 0.02 | 0.24 | 0.02 | 0.52 |
| 35 | Did you have a sense of mental agency, in terms of actively and deliberately thinking thoughts? | 0.28 | -0.09 | -0.14 | -0.03 | 0.05 | -0.04 | 0.28 | 0.02 | 0.05 | 0.03 | 0.24 | 0.07 | 0.49 |
| 17 | Was there pain? | 0.28 | -0.18 | 0.06 | -0.16 | 0.25 | -0.02 | 0.10 | 0.10 | 0.10 | 0.12 | 0.05 | 0.00 | 0.54 |
| 59 | Did you experience the space of pure awareness as having boundaries? | 0.27 | -0.20 | -0.16 | 0.00 | 0.18 | -0.07 | -0.04 | -0.24 | 0.03 | 0.14 | 0.01 | 0.04 | 0.49 |
| 57 | Did your experience have a quality that could be described as a non-physical "space"? | -0.23 | 0.15 | 0.19 | 0.00 | -0.01 | 0.19 | 0.04 | 0.21 | -0.06 | 0.04 | 0.07 | 0.04 | 0.59 |

*(Continued)*

**Table 6.** (Continued)

| Item Nr | Item long name | Factor 1 Time, Effort and Desire | Factor 2 Peace, Bliss and Silence | Factor 3 Self-Knowledge, Autonomous Cognizance and Insight | Factor 4 Wakeful Presence | Factor 5 Pure Awareness in Dream and Sleep | Factor 6 Luminosity | Factor 7 Thoughts and Feelings | Factor 8 Emptiness and Non-egoic Self-awareness | Factor 9 Sensory Perception in Body and Space | Factor 10 Touching World and Self | Factor 11 Mental Agency | Factor 12 Witness Consciousness | Uniqueness |
|---|---|---|---|---|---|---|---|---|---|---|---|---|---|---|
| 60 | Did you experience the space of pure awareness as divisible? | 0.22 | -0.17 | -0.15 | -0.08 | 0.17 | -0.02 | 0.01 | -0.11 | 0.02 | 0.22 | 0.00 | 0.05 | 0.60 |
| 29 | Was your experience part of your own inner life? | 0.20 | 0.15 | -0.01 | 0.16 | 0.15 | -0.01 | 0.10 | -0.18 | -0.12 | 0.11 | 0.15 | 0.07 | 0.76 |
| 71 | Did you feel relaxed? | -0.05 | 0.76 | 0.04 | -0.01 | -0.01 | 0.01 | -0.01 | -0.01 | -0.03 | -0.07 | 0.03 | 0.01 | 0.37 |
| 69 | Did you feel at peace? | -0.03 | 0.72 | -0.04 | 0.02 | -0.09 | 0.01 | -0.02 | 0.05 | -0.03 | -0.02 | 0.02 | -0.05 | 0.38 |
| 70 | Did you feel at ease? | -0.01 | 0.72 | 0.02 | 0.03 | -0.09 | 0.04 | -0.01 | -0.01 | -0.02 | -0.07 | 0.02 | -0.08 | 0.37 |
| 68 | Did you feel whole? | -0.14 | 0.54 | 0.13 | 0.09 | 0.00 | 0.02 | 0.04 | 0.02 | 0.01 | 0.07 | 0.02 | -0.07 | 0.44 |
| 65 | Was there a subtle but clearly positive mood, e.g. like an "invisible smile"? | -0.06 | 0.47 | -0.05 | 0.02 | -0.15 | 0.17 | 0.07 | -0.14 | 0.04 | 0.21 | 0.07 | 0.02 | 0.53 |
| 64 | Did your experience have a quality of "blissfulness"? | -0.02 | 0.45 | -0.09 | 0.11 | -0.10 | 0.23 | 0.21 | -0.02 | -0.02 | 0.18 | -0.05 | 0.00 | 0.49 |
| 67 | Did your experience have a quality of deep, unbounded silence? | -0.02 | 0.42 | 0.02 | 0.02 | 0.02 | 0.01 | -0.04 | 0.30 | -0.19 | 0.06 | -0.02 | 0.06 | 0.52 |
| 73 | Did you have a sense that the state you experienced was "the natural state"? | -0.03 | 0.40 | 0.34 | 0.03 | 0.03 | 0.05 | -0.03 | 0.08 | -0.03 | 0.03 | 0.05 | 0.02 | 0.53 |
| 63 | Did your experience have a quality of non-tactile and non-emotional "gentleness"? | 0.04 | 0.39 | 0.00 | -0.11 | -0.02 | 0.13 | -0.17 | 0.05 | -0.04 | 0.33 | 0.08 | 0.06 | 0.60 |

*(Continued)*

**Table 6.** (Continued)

| Item Nr | Item long name | Factor 1 Time, Effort and Desire | Factor 2 Peace, Bliss and Silence | Factor 3 Self-Knowledge, Autonomous Cognizance and Insight | Factor 4 Wakeful Presence | Factor 5 Pure Awareness in Dream and Sleep | Factor 6 Luminosity | Factor 7 Thoughts and Feelings | Factor 8 Emptiness and Non-egoic Self-awareness | Factor 9 Sensory Perception in Body and Space | Factor 10 Touching World and Self | Factor 11 Mental Agency | Factor 12 Witness Consciousness | Uniqueness |
|---|---|---|---|---|---|---|---|---|---|---|---|---|---|---|
| 55 | Did you have an experience of "pure being"? | -0.06 | 0.34 | 0.17 | 0.26 | 0.04 | 0.01 | -0.04 | 0.23 | 0.01 | 0.09 | -0.03 | -0.09 | 0.46 |
| 8 | Was your mood positive? | -0.05 | 0.32 | -0.17 | 0.28 | -0.12 | 0.18 | 0.21 | -0.06 | 0.00 | 0.08 | -0.02 | -0.12 | 0.58 |
| 86 | Is the experience of pure awareness the simplest kind of conscious experience you know? | 0.04 | 0.31 | 0.23 | -0.01 | 0.03 | -0.03 | -0.16 | 0.10 | 0.06 | 0.04 | 0.00 | 0.04 | 0.76 |
| 89 | Did your experience have a quality of non-acoustic "harmony" or "soundness"? | -0.13 | 0.29 | -0.01 | 0.08 | -0.02 | 0.09 | 0.08 | 0.05 | 0.00 | 0.29 | 0.00 | -0.20 | 0.60 |
| 52 | Did the experience have a quality of knowing itself? | 0.11 | 0.05 | 0.46 | 0.16 | -0.01 | 0.18 | 0.16 | 0.02 | -0.13 | 0.05 | 0.08 | 0.06 | 0.56 |
| 46 | Did your experience have a "cognizant" quality? | -0.10 | -0.06 | 0.42 | 0.19 | -0.07 | 0.14 | 0.20 | -0.04 | -0.11 | 0.13 | 0.13 | -0.11 | 0.54 |
| 44 | Did your experience have a quality of autonomy, meaning that it existed all by itself; that it came by itself and went by itself; that it was nothing you had created? | -0.19 | 0.05 | 0.42 | 0.01 | 0.00 | 0.14 | -0.08 | 0.03 | 0.07 | -0.06 | -0.18 | 0.06 | 0.59 |
| 75 | Was there a quality of insight? | -0.04 | 0.03 | 0.38 | 0.12 | -0.25 | 0.11 | 0.23 | 0.00 | -0.03 | 0.16 | 0.02 | 0.06 | 0.54 |

(Continued)

**Table 6.** (Continued)

| Item Nr | Item long name | Factor 1 Time, Effort and Desire | Factor 2 Peace, Bliss and Silence | Factor 3 Self-Knowledge, Autonomous Cognizance and Insight | Factor 4 Wakeful Presence | Factor 5 Pure Awareness in Dream and Sleep | Factor 6 Luminosity | Factor 7 Thoughts and Feelings | Factor 8 Emptiness and Non-egoic Self-awareness | Factor 9 Sensory Perception in Body and Space | Factor 10 Touching World and Self | Factor 11 Mental Agency | Factor 12 Witness Consciousness | Uniqueness |
|---|---|---|---|---|---|---|---|---|---|---|---|---|---|---|
| 54 | Would it be a good description to say that your experience of pure awareness, upon self-recognizing, also recognized itself as that awareness which had always been present in all experiences in the past? | 0.01 | 0.07 | 0.37 | 0.03 | -0.02 | 0.07 | -0.03 | 0.14 | -0.01 | 0.19 | 0.07 | 0.05 | 0.66 |
| 74 | Did you have a sense that the state you experienced was nothing you had fabricated? | -0.20 | 0.18 | 0.37 | -0.06 | -0.13 | 0.12 | -0.02 | 0.08 | 0.07 | -0.08 | -0.09 | 0.17 | 0.54 |
| 41 | Did your experience have a non-visual quality of "clarity"? | -0.08 | 0.11 | 0.31 | 0.23 | -0.06 | 0.26 | -0.08 | 0.06 | -0.01 | -0.06 | 0.06 | -0.08 | 0.49 |
| 45 | Was there an experience of "pure knowing" without any object? | 0.02 | 0.09 | 0.29 | 0.17 | 0.08 | 0.23 | -0.01 | 0.25 | -0.19 | -0.07 | 0.05 | -0.03 | 0.56 |
| 82 | Did you feel identical to the experience of pure awareness itself? | -0.07 | 0.19 | 0.26 | 0.19 | -0.03 | -0.04 | -0.01 | 0.20 | -0.01 | 0.12 | -0.02 | -0.11 | 0.59 |
| 77 | Were you looking not at pure awareness, but from pure awareness? | -0.07 | 0.22 | 0.25 | 0.07 | -0.08 | 0.01 | -0.10 | 0.16 | 0.04 | 0.10 | 0.06 | 0.03 | 0.64 |
| 21 | Did you have an experience of "now"? | 0.05 | 0.03 | 0.11 | 0.57 | -0.03 | -0.03 | -0.04 | -0.11 | -0.04 | 0.06 | 0.04 | 0.19 | 0.56 |

(*Continued*)

**Table 6.** (Continued)

| Item Nr | Item long name | Factor 1 Time, Effort and Desire | Factor 2 Peace, Bliss and Silence | Factor 3 Self-Knowledge, Autonomous Cognizance and Insight | Factor 4 Wakeful Presence | Factor 5 Pure Awareness in Dream and Sleep | Factor 6 Luminosity | Factor 7 Thoughts and Feelings | Factor 8 Emptiness and Non-egoic Self-awareness | Factor 9 Sensory Perception in Body and Space | Factor 10 Touching World and Self | Factor 11 Mental Agency | Factor 12 Witness Consciousness | Uniqueness |
|---|---|---|---|---|---|---|---|---|---|---|---|---|---|---|
| 20 | Did you have an experience of "here"? | 0.13 | 0.11 | 0.07 | 0.55 | 0.04 | −0.15 | −0.06 | −0.18 | 0.10 | 0.10 | −0.03 | 0.19 | 0.56 |
| 23 | Did you experience yourself as being in the present moment? | −0.19 | 0.14 | 0.10 | 0.51 | −0.04 | −0.02 | −0.05 | 0.04 | 0.04 | −0.04 | 0.05 | −0.05 | 0.47 |
| 2 | How alert were you? | −0.07 | −0.08 | −0.05 | 0.49 | −0.16 | 0.20 | −0.05 | 0.23 | 0.05 | −0.08 | 0.11 | −0.05 | 0.53 |
| 32 | Did you have the experience of simply "being", e.g. of simply being here and now? | −0.11 | 0.34 | 0.08 | 0.43 | 0.04 | −0.08 | −0.10 | 0.07 | 0.04 | −0.05 | 0.02 | −0.02 | 0.50 |
| 1 | How awake were you? | −0.02 | −0.11 | −0.01 | 0.43 | −0.19 | 0.26 | −0.09 | 0.22 | 0.02 | −0.07 | 0.08 | −0.07 | 0.55 |
| 15 | Was there an experience of oneness or unity? | −0.17 | 0.13 | 0.08 | 0.32 | −0.04 | 0.09 | 0.20 | 0.21 | −0.03 | 0.14 | −0.13 | −0.04 | 0.51 |
| 56 | Did you feel the fact of your own existence as such? | 0.07 | 0.16 | 0.24 | 0.24 | 0.09 | 0.00 | 0.01 | −0.16 | −0.09 | 0.14 | 0.11 | 0.00 | 0.72 |
| 83 | Did your experience occur during dreamless deep sleep? | −0.04 | −0.07 | 0.03 | 0.01 | 0.72 | 0.05 | −0.03 | 0.01 | 0.03 | 0.04 | −0.04 | 0.05 | 0.46 |
| 84 | Did your experience occur during the dream state? | −0.07 | −0.08 | −0.02 | 0.05 | 0.70 | 0.08 | 0.02 | −0.02 | 0.03 | 0.03 | −0.04 | 0.04 | 0.48 |
| 85 | Did your experience occur during the normal awake state, possibly together with thoughts and sensations? | 0.11 | 0.04 | 0.17 | 0.08 | −0.31 | −0.10 | 0.03 | 0.02 | 0.15 | 0.08 | 0.14 | −0.09 | 0.78 |

(Continued)

**Table 6.** (Continued)

| Item Nr | Item long name | Factor 1 Time, Effort and Desire | Factor 2 Peace, Bliss and Silence | Factor 3 Self-Knowledge, Autonomous Cognizance and Insight | Factor 4 Wakeful Presence | Factor 5 Pure Awareness in Dream and Sleep | Factor 6 Luminosity | Factor 7 Thoughts and Feelings | Factor 8 Emptiness and Non-egoic Self-awareness | Factor 9 Sensory Perception in Body and Space | Factor 10 Touching World and Self | Factor 11 Mental Agency | Factor 12 Witness Consciousness | Uniqueness |
|---|---|---|---|---|---|---|---|---|---|---|---|---|---|---|
| 40 | Did your experience have a non-visual quality of "brightness"? | 0.02 | -0.03 | 0.02 | -0.03 | 0.04 | 0.73 | -0.02 | 0.03 | 0.02 | -0.02 | 0.02 | -0.01 | 0.49 |
| 42 | Did your experience have a non-visual quality of "radiance" or "self-luminosity"? | 0.09 | 0.03 | 0.13 | 0.02 | 0.05 | 0.69 | -0.08 | -0.03 | 0.00 | 0.07 | -0.04 | -0.02 | 0.45 |
| 66 | Did your experience have a non-visual quality of being "radiant"? | 0.02 | 0.18 | 0.07 | -0.06 | -0.06 | 0.58 | -0.05 | -0.04 | 0.04 | 0.15 | -0.04 | 0.07 | 0.47 |
| 39 | Did you have a visual experience of brightness with closed eyes? | -0.06 | 0.04 | -0.18 | -0.05 | 0.20 | 0.53 | 0.26 | 0.04 | -0.02 | -0.05 | 0.00 | 0.08 | 0.57 |
| 62 | Did your experience have a quality that could be described as "vibrant", but in a non-tactile and non-visual sense? | -0.07 | 0.02 | 0.03 | 0.04 | -0.11 | 0.37 | 0.00 | -0.03 | 0.16 | 0.23 | 0.03 | 0.16 | 0.64 |
| 3 | Did you have thoughts? | 0.17 | -0.02 | 0.11 | -0.13 | 0.02 | -0.10 | 0.48 | -0.12 | 0.18 | -0.03 | 0.07 | 0.04 | 0.44 |
| 6 | Did you have mental images, e.g. motor imagery? | -0.10 | 0.01 | 0.03 | -0.06 | 0.22 | 0.05 | 0.48 | -0.07 | 0.22 | 0.01 | 0.01 | 0.05 | 0.56 |
| 7 | Did you have emotions? | 0.07 | 0.01 | -0.05 | 0.01 | -0.02 | 0.09 | 0.44 | -0.17 | 0.21 | 0.08 | -0.09 | -0.07 | 0.59 |
| 26 | Were there concepts in your mind? | 0.33 | -0.09 | 0.13 | -0.15 | -0.01 | -0.02 | 0.37 | -0.09 | 0.09 | -0.03 | 0.08 | 0.10 | 0.45 |
| 4 | Did you have spontaneously occurring memories? | 0.13 | 0.07 | 0.07 | -0.17 | 0.31 | -0.07 | 0.37 | 0.09 | 0.12 | 0.12 | 0.07 | 0.01 | 0.49 |

(*Continued*)

Table 6. (Continued)

| Item Nr | Item long name | Factor 1 Time, Effort and Desire | Factor 2 Peace, Bliss and Silence | Factor 3 Self-Knowledge, Autonomous Cognizance and Insight | Factor 4 Wakeful Presence | Factor 5 Pure Awareness in Dream and Sleep | Factor 6 Luminosity | Factor 7 Thoughts and Feelings | Factor 8 Emptiness and Non-egoic Self-awareness | Factor 9 Sensory Perception in Body and Space | Factor 10 Touching World and Self | Factor 11 Mental Agency | Factor 12 Witness Consciousness | Uniqueness |
|---|---|---|---|---|---|---|---|---|---|---|---|---|---|---|
| 5 | Were you planning or thinking about the future? | 0.23 | 0.02 | 0.07 | -0.17 | 0.31 | -0.06 | 0.35 | 0.05 | 0.08 | 0.06 | 0.04 | -0.05 | 0.48 |
| 27 | Was there discursive thought? | 0.28 | -0.05 | -0.08 | -0.23 | 0.08 | -0.05 | 0.32 | -0.01 | 0.04 | -0.04 | 0.16 | 0.11 | 0.41 |
| 43 | Did your experience vary in its intensity? | 0.15 | -0.01 | -0.21 | -0.12 | 0.03 | 0.01 | 0.25 | 0.06 | 0.08 | 0.02 | 0.13 | 0.12 | 0.68 |
| 13 | Could your experience be described as emptiness, a vacuum, or a void? | 0.06 | 0.12 | -0.10 | 0.03 | 0.11 | 0.00 | -0.07 | 0.45 | -0.21 | 0.09 | -0.05 | 0.16 | 0.63 |
| 53 | Would it be a good description to say that there was "an emptiness that has awoken to itself"? | 0.13 | 0.11 | 0.14 | 0.04 | 0.03 | -0.03 | -0.09 | 0.39 | -0.15 | 0.22 | -0.01 | 0.12 | 0.62 |
| 51 | Did you feel as though it was not you who had an experience of "pure knowing" without any object, it was rather as if the "pure knowing" was self-aware, knowing only itself, while you had nothing to do with it? | 0.00 | -0.04 | 0.25 | -0.05 | -0.05 | 0.12 | -0.13 | 0.38 | 0.04 | 0.07 | -0.11 | 0.18 | 0.59 |
| 28 | Was there a sense of self? | 0.24 | 0.08 | -0.06 | 0.14 | 0.20 | -0.04 | 0.24 | -0.37 | -0.06 | 0.00 | 0.14 | 0.01 | 0.49 |

(Continued)

**Table 6.** (Continued)

| Item Nr | Item long name | Factor 1 Time, Effort and Desire | Factor 2 Peace, Bliss and Silence | Factor 3 Self-Knowledge, Autonomous Cognizance and Insight | Factor 4 Wakeful Presence | Factor 5 Pure Awareness in Dream and Sleep | Factor 6 Luminosity | Factor 7 Thoughts and Feelings | Factor 8 Emptiness and Non-egoic Self-awareness | Factor 9 Sensory Perception in Body and Space | Factor 10 Touching World and Self | Factor 11 Mental Agency | Factor 12 Witness Consciousness | Uniqueness |
|---|---|---|---|---|---|---|---|---|---|---|---|---|---|---|
| 91 | Did you experience a dissolving of the boundary between yourself and your surroundings? For example, you no longer knew where our body started and ended, or you felt like you were either everything or nothing? | -0.25 | 0.08 | 0.03 | 0.01 | -0.12 | 0.01 | 0.16 | 0.36 | 0.00 | 0.14 | -0.17 | 0.14 | 0.57 |
| 92 | Did you have an experience of your breath stopping? E.g, in the sense that you felt you no longer had to breathe? | -0.12 | -0.03 | -0.11 | -0.04 | 0.18 | 0.07 | 0.04 | 0.28 | -0.12 | 0.15 | -0.02 | 0.19 | 0.74 |
| 72 | Did you experience the state of pure awareness as neither mental nor bodily? | -0.17 | 0.17 | 0.21 | -0.15 | 0.05 | 0.07 | -0.04 | 0.24 | -0.06 | 0.07 | -0.09 | 0.04 | 0.65 |
| 81 | Is it possible for you to put your experience of pure awareness into words? | 0.00 | 0.06 | 0.09 | 0.13 | 0.09 | -0.02 | 0.03 | -0.14 | 0.00 | 0.01 | 0.08 | 0.04 | 0.93 |
| 9 | Did you have perceptions, for example of objects in your environment? | 0.10 | 0.01 | 0.07 | 0.08 | -0.06 | -0.07 | -0.12 | 0.02 | 0.60 | 0.01 | 0.06 | 0.04 | 0.61 |
| 19 | Was there motion in space? | 0.12 | -0.07 | -0.01 | 0.08 | 0.10 | 0.10 | 0.03 | 0.08 | 0.53 | 0.04 | -0.02 | 0.04 | 0.59 |
| 18 | Did you have movement sensations? | 0.12 | -0.03 | -0.06 | 0.01 | 0.13 | 0.00 | 0.10 | -0.01 | 0.52 | 0.06 | 0.02 | 0.07 | 0.51 |

*(Continued)*

**Table 6.** (Continued)

| Item Nr | Item long name | Factor 1 Time, Effort and Desire | Factor 2 Peace, Bliss and Silence | Factor 3 Self-Knowledge, Autonomous Cognizance and Insight | Factor 4 Wakeful Presence | Factor 5 Pure Awareness in Dream and Sleep | Factor 6 Luminosity | Factor 7 Thoughts and Feelings | Factor 8 Emptiness and Non-egoic Self-awareness | Factor 9 Sensory Perception in Body and Space | Factor 10 Touching World and Self | Factor 11 Mental Agency | Factor 12 Witness Consciousness | Uniqueness |
|---|---|---|---|---|---|---|---|---|---|---|---|---|---|---|
| 16 | Were there sensory qualities, e.g. brightness, colour, sound, taste, smell? | -0.12 | -0.01 | -0.04 | 0.05 | 0.00 | 0.25 | 0.12 | -0.10 | 0.51 | -0.05 | 0.04 | 0.03 | 0.62 |
| 12 | Did you feel your body if you did not deliberately attend to it? | 0.25 | 0.03 | -0.01 | 0.10 | 0.01 | -0.11 | -0.07 | -0.15 | 0.37 | 0.09 | 0.10 | -0.02 | 0.63 |
| 11 | Did you have temperature sensations? | 0.14 | 0.07 | -0.08 | -0.09 | 0.16 | -0.02 | 0.14 | 0.07 | 0.35 | 0.12 | 0.03 | 0.02 | 0.63 |
| 14 | Was awareness itself the only content of awareness? | -0.11 | 0.06 | 0.00 | 0.19 | 0.15 | -0.03 | -0.21 | 0.18 | -0.30 | 0.04 | -0.04 | -0.05 | 0.62 |
| 10 | Did you have weight sensations? | 0.13 | 0.00 | -0.12 | -0.08 | 0.17 | -0.08 | 0.15 | 0.08 | 0.28 | 0.10 | 0.04 | 0.11 | 0.62 |
| 79 | Did your experience of pure awareness have an abstract quality of "self-touch", but not in a tactile sense? | -0.04 | -0.07 | 0.06 | 0.00 | 0.11 | 0.04 | -0.02 | -0.05 | -0.05 | 0.63 | 0.04 | 0.00 | 0.58 |
| 80 | Was your experience of pure awareness like the entire body touching, and simultaneously being touched by, the world? | -0.05 | -0.04 | -0.02 | 0.06 | 0.00 | -0.02 | 0.07 | 0.16 | 0.12 | 0.60 | -0.04 | -0.07 | 0.58 |

(Continued)

**Table 6.** (Continued)

| Item Nr | Item long name | Factor 1 Time, Effort and Desire | Factor 2 Peace, Bliss and Silence | Factor 3 Self-Knowledge, Autonomous Cognizance and Insight | Factor 4 Wakeful Presence | Factor 5 Pure Awareness in Dream and Sleep | Factor 6 Luminosity | Factor 7 Thoughts and Feelings | Factor 8 Emptiness and Non-egoic Self-awareness | Factor 9 Sensory Perception in Body and Space | Factor 10 Touching World and Self | Factor 11 Mental Agency | Factor 12 Witness Consciousness | Uniqueness |
|---|---|---|---|---|---|---|---|---|---|---|---|---|---|---|
| 78 | Did your experience of pure awareness resemble the concrete tactile experience you have when you touch your left hand with your right hand, creating the experience of touching and being touched at the same location simultaneously? | 0.06 | -0.13 | 0.04 | -0.09 | 0.02 | 0.01 | -0.09 | -0.02 | 0.05 | 0.54 | 0.02 | 0.10 | 0.66 |
| 61 | Did your experience have a quality that could be described as "velvety", but in a non-tactile sense? | 0.06 | 0.20 | -0.15 | -0.11 | 0.09 | 0.17 | -0.11 | 0.05 | 0.02 | 0.42 | 0.10 | 0.02 | 0.66 |
| 88 | Did your experience have a quality of non-tactile "density" or "fullness"? | -0.16 | 0.03 | 0.04 | 0.04 | 0.01 | 0.14 | -0.03 | -0.05 | 0.02 | 0.42 | 0.01 | 0.07 | 0.70 |
| 87 | Did you have an experience of pure awareness penetrating your body, e.g. like a field that also penetrates all other objects and living things? | -0.10 | 0.05 | 0.06 | 0.04 | -0.05 | 0.12 | 0.08 | 0.26 | 0.07 | 0.37 | 0.04 | -0.03 | 0.63 |
| 58 | Did you experience the space of pure awareness as having a centre? | 0.17 | -0.06 | -0.09 | 0.06 | 0.14 | 0.01 | 0.05 | -0.19 | -0.06 | 0.28 | 0.06 | 0.01 | 0.74 |
| 90 | Did you have a sense of awe or wonder? | -0.09 | 0.00 | 0.09 | 0.13 | -0.26 | 0.08 | 0.23 | -0.08 | 0.02 | 0.27 | -0.08 | 0.14 | 0.70 |

(Continued)

**Table 6.** (Continued)

| Item Nr | Item long name | Factor 1 Time, Effort and Desire | Factor 2 Peace, Bliss and Silence | Factor 3 Self-Knowledge, Autonomous Cognizance and Insight | Factor 4 Wakeful Presence | Factor 5 Pure Awareness in Dream and Sleep | Factor 6 Luminosity | Factor 7 Thoughts and Feelings | Factor 8 Emptiness and Non-egoic Self-awareness | Factor 9 Sensory Perception in Body and Space | Factor 10 Touching World and Self | Factor 11 Mental Agency | Factor 12 Witness Consciousness | Uniqueness |
|---|---|---|---|---|---|---|---|---|---|---|---|---|---|---|
| 50 | Did you have a sense of confidence that you would be able to know things in the future? | 0.05 | -0.09 | 0.12 | 0.02 | 0.15 | 0.14 | 0.19 | -0.04 | -0.03 | 0.24 | 0.16 | -0.01 | 0.73 |
| 76 | Was your experience of pure awareness different from an experience of pure awareness that "has become aware of itself"? | -0.03 | -0.07 | -0.10 | 0.01 | 0.05 | 0.02 | 0.04 | 0.01 | -0.03 | 0.12 | -0.02 | -0.03 | 0.97 |
| 47 | Did you know that you would be able to deliberately control your attention if you wanted to? | -0.10 | 0.04 | 0.00 | -0.04 | -0.03 | 0.01 | -0.05 | 0.01 | 0.02 | -0.01 | 0.81 | 0.03 | 0.38 |
| 48 | Did you know that you would be able to deliberately think thoughts if you wanted to? | -0.02 | 0.02 | 0.05 | -0.01 | -0.01 | -0.02 | -0.05 | 0.04 | 0.04 | -0.01 | 0.78 | -0.01 | 0.41 |
| 49 | Did you have a sense of confidence that you would still be alert in the next moment? | -0.06 | 0.05 | 0.13 | 0.10 | -0.06 | 0.07 | -0.03 | -0.11 | -0.08 | 0.05 | 0.47 | -0.02 | 0.68 |
| 37 | Were you aware of meditating? | 0.07 | -0.01 | -0.15 | 0.15 | 0.00 | -0.05 | 0.13 | -0.07 | -0.14 | 0.04 | 0.41 | 0.06 | 0.70 |
| 34 | Did you have a sense of mental agency, in terms of actively and deliberately controlling the focus of your attention? | 0.17 | -0.03 | -0.24 | 0.08 | 0.00 | -0.05 | 0.23 | 0.01 | 0.04 | 0.02 | 0.39 | 0.11 | 0.52 |

(Continued)

**Table 6.** (Continued)

| Item Nr | Item long name | Factor 1 Time, Effort and Desire | Factor 2 Peace, Bliss and Silence | Factor 3 Self-Knowledge, Autonomous Cognizance and Insight | Factor 4 Wakeful Presence | Factor 5 Pure Awareness in Dream and Sleep | Factor 6 Luminosity | Factor 7 Thoughts and Feelings | Factor 8 Emptiness and Non-egoic Self-awareness | Factor 9 Sensory Perception in Body and Space | Factor 10 Touching World and Self | Factor 11 Mental Agency | Factor 12 Witness Consciousness | Uniqueness |
|---|---|---|---|---|---|---|---|---|---|---|---|---|---|---|
| 31 | Did you have the experience of a passive observer being present? | -0.02 | 0.00 | -0.01 | 0.05 | 0.02 | -0.01 | 0.00 | -0.01 | 0.00 | -0.05 | 0.03 | 0.78 | 0.41 |
| 30 | Did you have the experience of an impersonal observer being present? | -0.01 | -0.05 | 0.02 | 0.04 | -0.02 | 0.03 | -0.02 | 0.05 | 0.02 | 0.00 | 0.01 | 0.75 | 0.43 |

**Table 7. Factor correlations[a].**

| | Time, Effort and Desire | Peace, Bliss and Silence | Self-Knowledge, Autonomous Cognizance and Insight | Wakeful Presence | Pure Awareness in Dream and Sleep | Luminosity | Thoughts and Feelings | Emptiness and Non-egoic Self-awareness | Sensory Perception in Body and Space | Touching World and Self | Mental Agency | Witness Consciousness |
|---|---|---|---|---|---|---|---|---|---|---|---|---|
| Time, Effort and Desire | 1.00 | | | | | | | | | | | |
| Peace, Bliss and Silence | **-0.40** | 1.00 | | | | | | | | | | |
| Self-Knowledge, Autonomous Cognizance and Insight | -0.26 | **0.32** | 1.00 | | | | | | | | | |
| Wakeful Presence | -0.22 | **0.34** | 0.28 | 1.00 | | | | | | | | |
| Pure Awareness in Dream and Sleep | **0.35** | -0.29 | -0.20 | -0.29 | 1.00 | | | | | | | |
| Luminosity | -0.24 | **0.30** | 0.24 | 0.19 | -0.08 | 1.00 | | | | | | |
| Thoughts and Feelings | **0.31** | -0.06 | -0.10 | -0.12 | 0.23 | 0.10 | 1.00 | | | | | |
| Emptiness and Non-egoic Self-awareness | **-0.30** | 0.21 | 0.22 | 0.07 | -0.07 | 0.23 | -0.19 | 1.00 | | | | |
| Sensory Perception in Body and Space | **0.37** | -0.19 | -0.08 | -0.08 | 0.11 | 0.01 | 0.28 | -0.19 | 1.00 | | | |
| Touching World and Self | 0.06 | 0.24 | 0.15 | 0.12 | 0.15 | **0.31** | 0.18 | 0.13 | 0.09 | 1.00 | | |
| Mental Agency | 0.28 | 0.05 | -0.03 | 0.11 | 0.00 | 0.01 | 0.20 | -0.15 | 0.12 | 0.10 | 1.00 | |
| Witness Consciousness | 0.25 | -0.13 | 0.03 | -0.02 | 0.22 | 0.07 | 0.09 | 0.07 | 0.15 | 0.19 | 0.10 | 1.00 |

[a]Correlations ≥ |0.3| are printed in boldface.

"wholeness" show an absence of mental conflict and an increased degree of integration. This is to be expected in all states in which a) the constant competition of different mental processes for the focus of attention (for example, the automatic arising of spontaneous, task-unrelated thought; [23–26]) has subsided, and b) representational contraction into a first-person perspective [27] and the resulting fragmentation of one's overall experiential space by goal-directed mental agency have been attenuated.

A low degree of perturbation also confirms the second of six semantic constraints for the working concept of "MPE" which were extracted in the above-mentioned study, this one termed "Low Complexity" (PC2; cf. [9]). Generally speaking, PC2 is often described as the complete absence of intentional content, in particular of high-level symbolic mental content (i.e., discursive, conceptual or propositional thought), but sometimes even as the disappearance of all sensorimotor, interoceptive and affective content. There is a weak and a strong reading of this semantic constraint, distinguishing between the mere phenomenological absence of cognitive agency, mind-wandering or mind blanking on the one hand (for details,

see [9]) and a "full-absorption episode" (as defined above) on the other. Both readings are compatible with **Factor 2**.

Recent research on mind-wandering has shown that "a wandering mind is an unhappy mind" [28]. Three items in **Factor 2** pick out positive mood and the phenomenal quality of "bliss" as often co-emerging with the silent mind of the meditator. A preliminary qualitative analysis of 841 phenomenological reports indicates that MPE *as such* is not an emotional state, but that it can certainly trigger a whole spectrum of mostly positive affective states like joy, existential relief, gratitude, unpersonal love, awe and wonder (but see [29]). In particular, MPE can sometimes co-exist with a mostly subtle but clearly noticeable form of bliss, an experience that has sometimes been described as an "invisible smile" (Item 65). **Factor 2** therefore expresses not only the phenomenology of peace, existential ease and silence, but also various forms of what in German is called *Stilles Entzücken* ("silent delight"), which seems intimately connected to a calm and entirely undramatic phenomenology of rapture and "non-sensational awe."

**Factor 3** was termed "Self-Knowledge, Autonomous Cognizance and Insight". It matches one of the two most central semantic constraints for the concept of "MPE", namely "Epistemicity" (PC5 in [9]), the non-conceptual phenomenal experience of "knowingness." This means that MPE instantiates an autonomous (i.e., unfabricated) phenomenal character of insight, cognizance and clarity.

The strongest-loading item in **Factor 3** ("Did the experience have a quality of knowing itself?") expresses a phenomenology of non-egoic, first-order reflexivity: The non-conceptual quality of "pure knowing" is *self-directed*, but in a non-egoic way, without the involvement of any kind of mental or bodily agency, without a conscious sense of control or ownership, and excluding the phenomenology of "selfhood" in terms of transtemporal identity. Accordingly, "pure knowing" also lacks the phenomenal experience of personhood, a conscious representation of being a rational individual possessing specific personality traits or any form of autobiographical narrative. Phenomenologically, pure awareness simply knows itself, timelessly. The interesting discovery is exactly that there is now empirical evidence for a non-egoic, homunculus-free form of self-awareness. Therefore, we also find a strong phenomenological relationship to **Factor 8** ("Emptiness and Non-egoic Self-awareness"). Our own approach is one of evidence-based phenomenology, but in ancient contemplative traditions this type of state has for many centuries been described epistemically, as "self-knowing timeless awareness" or as "self-cognizing wakefulness" (for example, in the Tibetan notion of *rang rig ye shes*).

**Factor 4** ("Wakeful Presence") integrates the phenomenology of spatiotemporal self-location (cf. [30]) with alertness and the unified experience of "existence as such." There is a phenomenal experience of being fully settled in the "Here" and the "Now," permeated by the character of wakefulness and a feeling of "simply being." We labelled this cluster of phenomenal qualities "Wakeful Presence" because its four strongest-loading items refer to the embodied experience of wakefully being in the present moment. The alertness component directly relates to and confirms the first, and most prominent, semantic constraint for the concept of "MPE," as derived from a previous study, namely "Wakefulness" (PC1 in [9]). If one interprets wakefulness as a functionally autonomous conscious representation of *epistemic capacity* (for example, of the existing capacity for self-orientation in time and space, plus the capacity for attentional control), then one finds a direct link to **Factor 3** ("Self-Knowledge, Autonomous Cognizance and Insight"). In Tibetan Buddhism, the relevant phenomenology was described more than a thousand years ago as "self-generating ever-fresh awareness" ([31], xi) and as the experiential quality of originary, naturally present and non-transient "primordial knowing" (*ye shes* in Tibetan; cf. [32], p447, [33], p99).

Factor 4 also incorporates two additional elements, however, namely items pointing to the phenomenal qualities of unity and of onticity (pure being, or being *as such*). The latter has also been reported in the context of syncopes and accidents (for discussion, see [9], note 21). Both reappear in our qualitative assessment under the rubric of "non-dual being."

**Factor 5** ("Pure Awareness in Dream and Sleep") is a special case and will, if replicated, be treated more thoroughly in future publications. It relates to the experience of pure awareness during NREM (non-rapid eye movement) sleep, which is a type of full absorption [34, 35]. As the second "stand-alone instance" of our research target, it will be of great relevance in triangulating the neural correlates of MPE. A number of first-person reports from our study support the existence of such states.

Pure awareness during NREM-sleep has previously been termed "lucid dreamless sleep" or (by the TM movement) "witnessing sleep." **Factor 5** shows interesting relations not only to the phenomenologies of "wakeful presence" and "non-dual being" already mentioned, but also to **Factor 12** ("Witness Consciousness") and **Factor 6** ("Luminosity").

**Factor 6** ("Luminosity") refers to non-visual phenomenal qualities of "brightness," "radiance" and "vibrancy," but also to the visual experience of brightness with closed eyes. In addition, "Self-Luminosity" (PC3) was one of the six semantic constraints previously extracted, where it may also be found as the phenomenology of "brilliance" or as the "clear light of primordial awareness."

The specific kind of phenomenal character sometimes described as "luminosity," "radiance" or even "enlightenment" comes in different varieties. For example, many practitioners describe a non-visual phenomenology of "clear light," a non-perceptual experience of clarity and mere epistemic openness, while others report a more concrete form of visual brightness, which can be experienced with closed as well as with open eyes.

According to classical Buddhist teachings, the term "luminosity" simply refers to an entirely non-conceptual experience of *epistemic capacity*, the capacity to know and experience (Tib. *salwa*; e.g., [36], p36; n. 30), and would therefore be directly related to **Factor 3** ("Self-Knowledge, Autonomous Cognizance and Insight"). Here, we could also speak of the phenomenal experience of *epistemic clarity*. This involves an open, currently unobstructed inner *space of knowing*, a space in which orientation and perceptual processes can unfold, in which attention can be controlled and focused or in which concepts can be formed and applied to experience. MPE would be a model of this space.

**Factor 7** ("Thoughts and Feelings") describes a classical meditation experience, just like the phenomenal descriptors picked out by "Time, Effort and Desire" (**Factor 1**). Thoughts of different kinds, mental images, emotions, memories and simulations of future events spontaneously arise. There is time experience, and the space of silent, spacious awareness into which the meditator wants to settle is more or less clouded. Phenomenologically, there is a mostly noisy foreground of active mental content.

**Factor 8** ("Emptiness and Non-egoic Self-awareness") may be most interesting from a philosophical perspective. In our study, it refers to an experiential quality of "pure knowing" without a sense of self and without any object. The relevant items describe an uncontracted, unbounded, and non-dual experience of wakefulness without content, which is self-aware, knowing only itself in a way that lacks any markers of egoic self-consciousness like agency, ownership, and autobiographical narrative, or spatiotemporal self-location [27, 30]. However, our formulation of the strongest-loading item in "Emptiness and Non-egoic Self-awareness" may mistakenly conflate "emptiness" with the merely spatial phenomenology of consciously experiencing a void or a vacuum.

"Emptiness" (*suññatā* in Pali) is one of the foremost concepts in Buddhist philosophy. It has been discussed by scholars and practitioners for far more than two thousand years. From a

*metaphysical* perspective, "emptiness" means that all phenomena lack substantiality and an intrinsic nature of their own. From a *phenomenological* perspective, full-absorption states in which pure awareness remains as the only reportable phenomenal character are a prime candidate for an "experience" of emptiness, the stand-alone quality of epistemic openness as discussed in the context of **Factor 6** above. Phenomenologically, "seeing the empty nature of phenomena" can also refer to a conscious experience without the slightest trace of conceptual overlay, namely to the distinct and crystal-clear phenomenology of seeing and perceiving out of timeless silence.

Interestingly, the second- and third-strongest loading items in **Factor 8** were those offering two metaphorical descriptions of first-order reflexivity combining the "Self-Knowledge, Autonomous Cognizance and Insight" of **Factor 3** and the "Wakefulness" of **Factor 4** with the phenomenal quality of emptiness and epistemic openness. These are negatively correlated with the phenomenology of selfhood: The phenomenology of self-knowing and self-awakening picked out by this factor is *non-egoic*. Arguably, it is exactly these aspects of **Factor 8** that, for many participants, may express the "spiritual essence" of MPE most directly. Evidence for the actual existence of non-egoic self-awareness in the context of a substantial psychometric study is a theoretically relevant result.

**Factor 9** ("Sensory Perception in Body and Space") describes the presence of sensory qualities, movement sensations and conscious body experience, implying that the phenomenal character of MPE can co-emerge with perceptual content. Qualitative analysis of reports shows it as related to the frequently described phenomenology of "direct perception." For many centuries, meditators have reported states of direct perception, the experience of seeing *what is*. Seeing *what is* out of a state of pure awareness often reveals another particular and interesting, phenomenal quality, namely the experience of "suchness" or "thusness", the ineffable uniqueness and particularity of any individual instance of non-conceptual content. This suggests another evidence-based, phenomenologically grounded reading of the "purity" of pure awareness, not as the absence of perceptual content, but as a complete lack of conceptual overlay and cognitive penetration, including time experience and judgements as to the "existence" or "non-existence" of what is perceived.

**Factor 10** ("Touching World and Self") describes MPE as an abstract form of tactile experience resembling self-touch (for example, the specific sensation of inter-manual self-touch) or as a tactile experience in which the entire body touching the world while simultaneously being touched by it. Other phenomenological descriptors are "velvety," "dense" and "full"—but always in a non-tactile way, i.e., as lacking the concrete phenomenal qualities normally characterizing the stimulus-correlated sense of touch. Here, the phenomenological profile of MPE is described, first, as an abstract form of *contact* or of "being in touch;" second, it can exhibit the phenomenal character of reflexivity (as in self-touch; or **Factor 8**; [37, 38]); and, third, it can be globalized, namely in the form of "an experience of pure awareness penetrating your body, e.g. like a field that also penetrates all other objects and living things" (Item 87). One interesting detail is the correlation of **Factor 10** with **Factor 6** ("Luminosity"). As there is no obvious phenomenological relationship between qualities like "brightness," "brilliance" or "radiance" and the ones just mentioned, this might perhaps point to interesting commonalities in the neural substrate.

**Factor 11** ("Mental Agency") points to subtle forms of egoic mental self-awareness, like being aware of the fact that one is currently meditating, experiencing the *potential* for sustained alertness and cognitive-attentional self-control, or even—in Item 34—having an explicit sense of mental agency unaccompanied by any *actually* ongoing, deliberate control of the focus of attention. This factor picks out phenomenological configurations in which egoic awareness, specifically the more subtle phenomenal qualities emerging from representing

one's own capacity for mental action, are still superimposed onto MPE. This feature also refers back to the notion of "dual mindfulness" as introduced when discussing **Factor 1** ("Time, Effort and Desire").

Finally, **Factor 12** ("Witness Consciousness") demonstrates how a classical term from the Advaita Vedanta (i.e., *sākṣin*; Sanskrit: साक्षी; [39, 40]) system of philosophy finds its expression on the level of phenomenology. Witness consciousness is that which makes all knowledge possible, cannot itself become an object of knowledge and is self-luminous (for more, see [9]). "Witness consciousness" also is a global phenomenology. This is to say that the *totality* of all experiential contents is experienced as being observed by something that isn't really an "observer" at all, but rather a timeless, absolutely impersonal, knowing presence. Phenomenologically, the world is mirrored in an all-encompassing quality of infinite, choiceless and non-conceptual knowing.

It is interesting to note how **Factors 3** ("Self-Knowledge, Autonomous Cognizance and Insight") and **4** ("Wakeful Presence") offer *phenomenological* support for results from the text-based evidence synthesis presented in [9], in which the two most important *semantic* constraints for MPE emerge as and "Epistemicity" (PC3) and "Wakefulness" (PC1), while **Factor 6** (Luminosity") directly maps onto the constraint which was termed "self-luminosity" (PC3) in this earlier investigation. Meanwhile, a highly intuitive pattern found in the present data is that the qualities of control, desire, effort and time experience found under **Factor 1** are clearly anti-correlated with "Peace, Bliss, and Silence (**Factor 2**) and what is arguably the most "spiritual" factor (**Factor 8**, "Emptiness and Non-egoic Self-awareness"), while positively correlating with dream, sleep, and mental activity on the cognitive and perceptual level.

Insofar as it is phenomenologically plausible, the 12-factor solution presented shows that a dimensional approach is viable and that the concept of "MPE" may be a graded construct which refers to a multidimensional space of possible conscious states, a space which may contain different points, regions or trajectories [9, 41]. MPE might then be a cluster concept with a probabilistic rather than a definitional structure, where membership is graded along multiple dimensions and some phenomenological exemplars are more prototypical than others. Since "pure awareness" experiences may be maximally prototypical, we might conjecture that a majority of respondents in our study will describe such states as the *simplest* kind of conscious experience they know. This is borne out by the fact that the corresponding item #86 in our questionnaire achieved a median rating of 80. However, it is important not to conflate subjective ratings of "simplicity" with whatever objective measures we may develop in the future. Accordingly, future research may demonstrate that even less complex forms of phenomenal character do exist and constitute the simplest form of conscious experience neurotypical human beings are capable of.

## Limitations

Our study has a number of limitations. Most fundamentally, there is no guarantee that participants understood the instructions and the concept of "pure awareness" introduced therein in the way we intended them. Varying understandings may have led to unwanted variation in responses. Even assuming a singular, precise and unequivocal understanding, it is still possible that some responses were driven largely by a desire to report particularly impressive or personally meaningful experiences rather than to adhere as closely as possible to the instructions. There is some evidence for this in the 841 phenomenal reports collected as part of the questionnaire, where a number of participants chose to report not "ordinary" MPE as it may occur during formal meditation practice and full-absorption episodes, but rarer and often quite dramatic non-dual awareness states in which all subject–object structure had spontaneously

disappeared. This may have been an indirect effect of us asking participants for a description of an experience "in which the quality of pure awareness was particularly salient and/or one which you can remember particularly clearly." What can be remembered particularly clearly may often be the most impressive states, which may not necessarily be the most paradigmatic instances of "pure awareness."

More generally, differences in understanding of *any* part of the survey (items, demographic questions) could have led to undesired response variability. Such differences can occur for a number of reasons. First, participants will typically have been exposed to different background information about meditation. If they have actively engaged with the relevant literature, there may have been large differences in what was selected for study. For instance, many of our participants (77%) were regular practitioners who had been meditating for years. It is plausible to assume that such a consistent habit may often be anchored in adherence to specific belief systems and the conceptual framework of a certain lineage or spiritual tradition or a specific teacher or organization. The terminology employed by such theories, as well their epistemological and metaphysical background assumptions, may "contaminate" survey responses (and phenomenological reports). We know from correspondence that such belief systems can sometimes even lead to a decision not to participate in scientific projects or to reject any attempt to try to approximate something considered to be as fundamentally ineffable and soteriologically relevant as MPE. Therefore, "theory contamination" may have both biased questionnaire responses and introduced a selection bias into the sample.

A second issue is what could be called "response drift": Some participants may have started out with a somewhat vague idea of the target state of pure awareness, which became increasingly focused as they moved through the questionnaire. Conversely, they may have started out with a very specific concept, perhaps dictated by commitment to a particular theory, and then similarly shifted as they moved through the questions. Responses given earlier may therefore have related to a somewhat different target than what later responses referred to. This could explain the observed divergence in responses to the two control items, which occurred at different places in the questionnaire, but were nearly identically phrased. In hindsight, it might have been better to phrase these items not just nearly, but absolutely, identically in order to be sure that any intra-individual variation in responses must be due to inconsistent reporting. As it was, divergent scores could also be due to the slight semantic difference in the two items, one asking about non-visual radiance only, while the other also asked about self-luminosity. We did not exclude any participants based on the difference in their control item scores. We are generally very cautious about excluding participants and prefer to err on the side of inclusion unless there is a compelling rationale for doing otherwise. Also, a non-arbitrary exclusion criterion is hard to find in this case. But most importantly, a single pair of control items is much too noisy a criterion to offer a sound basis for data exclusion.

In studies of "private" subjective experiences, there is a general set of problems associated with verbal report. One issue is ineffability: the difficulty or impossibility of expressing certain phenomenal states or qualities in words. This problem may be particularly pronounced when it comes to MPE, since the experiential quality of "pure awareness" has long been regarded by meditators as the paradigmatic example of ineffability. The ineffability problem may arise for several reasons. One is a lack of *concurrent reportability* during full-absorption episodes ([9], p14). Due to the non-dual nature of such states (i.e., the lack of any phenomenally represented subject–object structure) there will be no intention or cognitive capacity to verbally report, mentally categorize or actively memorize the phenomenal character in question. Another reason may be that MPE's timeless content is *unlike* [42, 43] any of the more familiar sensory, motor and interoceptive qualities, making it hard to grasp it verbally by comparing it to such qualities. A final source of ineffability is that the experience of MPE typically seems to lack any

internal structure or "grain," It seems uniformly dense. This phenomenal quality of "ultra-smoothness" [44–46], as it is called in the literature, may further hinder verbal report by depriving it of any discernible entry points for description or functional analysis.

Alongside ineffability issues, another problem faced by all studies based on retrospective self-report is the unreliability of memory recall. Human autobiographical memory, including recall of previous mental states, is notoriously fallible (for a review see [47]). In our study, we asked participants to report on experiences that occurred in their past, possibly even decades ago. Memory retrieval for such events will be necessarily biased, depending on various factors such as the amount of time that has since passed. The uncertainty introduced by such errors may be substantial but is likely to remain unavoidable in studies of this kind.

On a conceptual level there is an even deeper variant of the memory problem: If the concept of "autobiographical memory" refers to the representation of events which at the time at which they occurred were phenomenally represented as being experienced by a conscious *self*, then a participant's claims that *they themselves* have experienced a selfless state are highly dubious from a methodological perspective, because they contain either a logical or a performative fallacy ([48], p566). A memory of a pure-awareness experience could not be *auto*biographical memory in this strong phenomenological sense. Of course, it may be empirically possible that selfless states are stored and later retrieved in an autobiographical format, as a *post hoc* mnemonic misrepresentation adding the feature of egoic self-awareness–a case of misremembering, and not of confabulation (cf. [49, 50] for important discussions). But the relevant evidence has yet to be gathered.

One weakness of our anonymized online survey approach is that the identities of participants could not be verified, nor could we detect fraudulent responses or multiple responses by the same person. As part of privacy protection, IP addresses were not stored and could therefore not be used for data plausibility checks.

Finally, our data are highly unlikely to be representative of the global population of meditators. The online nature of the questionnaire and its distribution channels as well as self-selection will have produced a sample that deviates in several respects from the global population. The high number of incomplete responses (over 60%) also indicates a massive self-selection process, probably driven in part by the large number of questionnaire items, which may have been off-putting. However, representativeness is not a major concern at this stage. Future versions of this questionnaire will be put to the test against different populations and will be modified to provide a factor structure that holds up in as many settings as possible. Methods from confirmatory factor analysis can then be used to quantify the differences in item functioning and response patterns among different groups of participants.

## Conclusion

A central notion in traditional philosophical theories of meditation is that of "pure consciousness" or "pure awareness," the experience of being non-conceptually aware of consciousness *itself*. Based on responses to an online questionnaire, we here offer a factor structure to map the phenomenal character of such experiences in a fine-grained way. The 12 resulting dimensions not only reproduce important aspects of this phenomenon as reported in the literature, such as the actual existence of non-egoic self-awareness, they also point to some potentially new connections such as that between "Luminosity" and the abstract, non-tactile experience of self-touch.

However, our factor solution accounts for less than half of the item variance and quantitative measures indicate a lack of fit, particularly when simple structure is imposed. Future studies will be needed to improve upon this first version of the rating scale and to continue to

validate it in different populations and settings, including sleep states, non-meditative states and altered states of consciousness.

The under-researched experience of "pure awareness" in meditation is of great relevance to the interdisciplinary field of consciousness research. Pure awareness opens up a new methodological perspective by generating potential for a minimal model explanation, which in turn will contribute to the construction of a minimal unifying model (cf. [9, 51]). The study of pure awareness also helps in refining important theoretical questions in consciousness studies, for instance as regards ineffability and the relevance of dimensional approaches to fine-grained phenomenological analysis. Finally, it may foster theoretical unification by connecting the major existing approaches in a new way.

## Acknowledgments

We thank Tiziano Furlanetto, Solène Neyret, Adriana Alcaraz Sanchez, Jennifer Windt and Raphaël Millière for help with questionnaire translation. We are grateful to Tiziano Furlanetto, Solène Neyret, Adriana Alcaraz Sanchez and Wanja Wiese for help with translation and qualitative assessment of phenomenological reports. In particular, TM wants to thank all those committed practitioners who supported the project during the pilot and distribution phase, including Nicole Baden, Tilman Borghardt, Irene Bumbacher, Cyril Costines, Safae Essafi, Nicole Fasel, Catherine Felder, Matt Gwyther, Sam Harris, Britta Hölzel, Günter Hudasch, Yuka Nakamura, Muho Nölke, Ulrich Ott, Fred von Allmen, Toby Woods and others. We are greatly indebted to Erich Studerus for comments, and to Emily Troscianko, Wanja Wiese, and Cyril Costines for editorial help.

## Author Contributions

**Conceptualization:** Alex Gamma, Thomas Metzinger.

**Data curation:** Alex Gamma.

**Formal analysis:** Alex Gamma.

**Funding acquisition:** Thomas Metzinger.

**Methodology:** Alex Gamma, Thomas Metzinger.

**Resources:** Thomas Metzinger.

**Supervision:** Thomas Metzinger.

**Visualization:** Alex Gamma.

**Writing – original draft:** Alex Gamma, Thomas Metzinger.

**Writing – review & editing:** Alex Gamma, Thomas Metzinger.

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
