## [Decision Letter · Decision Letter 0]

13 Apr 2021

PONE-D-20-34727

The Minimal Phenomenal Experience questionnaire (MPE-92M): Towards a phenomenological profile of “pure awareness”-experiences in meditators

PLOS ONE

Dear Dr. Gamma,

Thank you for submitting your manuscript to PLOS ONE. I am very sorry it has taken such a long time to review your paper - we have had great difficulty in securing reviewers. The two reviewers are very positive about your paper but have recommended some minor revisions. Therefore, we invite you to submit a revised version of the manuscript that addresses the points raised during the review process.

We look forward to receiving your revised manuscript.

Kind regards,

Jane Elizabeth Aspell, PhD

Academic Editor

PLOS ONE

Journal Requirements:

3. Please change "female” or "male" to "woman” or "man" as appropriate, when used as a noun (see for instance https://apastyle.apa.org/style-grammar-guidelines/bias-free-language/gender ).

Reviewers' comments:

Reviewer's Responses to Questions

**Comments to the Author**

1. Is the manuscript technically sound, and do the data support the conclusions?

Reviewer #1: Yes

Reviewer #2: Yes

2. Has the statistical analysis been performed appropriately and rigorously? 

Reviewer #1: Yes

Reviewer #2: Yes

3. Have the authors made all data underlying the findings in their manuscript fully available?

Reviewer #1: Yes

Reviewer #2: No

4. Is the manuscript presented in an intelligible fashion and written in standard English?

Reviewer #1: Yes

Reviewer #2: Yes

5. Review Comments to the Author

Reviewer #1: 1. Summary of the research and impression

The study aims to explore states of ‘Pure Consciousness’ by conducting a large questionnaire study into the phenomenology of such experiences. The area has deep cultural roots in contemplative traditions which the researchers are keen to acknowledge. Their approach is to cast a wide net and examine the phenomenology of such experiences agnostic to tradition and in recognition of a greater interest from Western secular approaches over recent years. Research to date has been descriptive and focussed on phenomenology. This study takes a psychometric approach informed by the theory of Minimal Phenomenological Experience (MPE). The questionnaire is based on research and contemplative literature and previous pilot studies. There were 92 items which the researchers hoped to cluster into dimensions. The researchers conducted exploratory factor analysis and chose a 12-factor solution providing sufficient factor loading (0.3) and meeting conceptual criteria. Problems with the chosen factor solution are acknowledged at the start of discussion which limit its psychometric contribution. The discussion provides an illuminating theoretical interpretation of the findings in the context of previous work on the MPE project. It establishes the theoretical contribution of this piece in further explaining more fine-grained elements of pure consciousness and their interrelationships while situating discussions in the context of traditional contemplative literature. There are novel lines of investigation suggested that will help further illuminate the concept and phenomenology of Pure Consciousness and the wider MPE endeavours such as the link between ‘Touching world and Self’ and ‘Luminosity’.

The paper is clearly written and is a unique and theoretically compelling contribution to the literature. Shortfalls in model fit from the questionnaire items are a shame and somewhat limit the psychometric contribution of this work. Having said this, the robust attempts are certainly necessary and go some way to teasing out phenomenological distinctions which are well explored in the discussion and will prove fruitful for researchers attempting to narrow down hypotheses and explore experimental / psychometric approaches. Shortfalls are well acknowledged and discussed in detail in the limitations section, including what feel like the most salient points around theory contamination, terminological understanding, ineffability and autobiographical memory recall. The second paper on phenomenological reports is much anticipated and will be a useful companion to the discussion here which builds on the theoretical work of the MPE project. Overall, this is an extremely interesting and necessary piece of work, its scale, ambition and interdisciplinary approach provide a significant contribution to our understanding of these multivarious and compelling experiences.

2. Discussion of specific areas for improvement

I recommend this paper for publication with some minor revisions (a) which may strengthen the paper by making it clearer for a general readership and (b) help tease out some of the limitations to aid future researchers building on this work:

• 117: Minimalist idealization – terminology is not clear to general readership and non-philosophers, perhaps define

• 116 – 125: rationale for this second goal (minimal model explanation for conscious experience) is clearer in MPE paper (ref 10) – worth considering adding note on parsimony of such approach to clarify why this is a theoretical target

• 180: Total usable responses were just 38.7% - the ‘Participants’ section (168) outlines minimum requirements, but doesn’t comment on why? Is there a limitation here around survey length and why there were so many incomplete responses? Consider adding to discussion of response drift (907)

• 210: typo – I think this should say table 6

• 248: does not mention whether control items (42 & 66) indicated reliable responses. This is discussed at 424 – although the authors do not suggest whether there was a criterion to exclude respondents based on responses. i.e. if there was a cut off for differences on control items over some arbitrary number. This is discussed in Limitations (914) and reasons for differences are mentioned with the salient point being that items were not identical. However, it is not acknowledged why no data was excluded on basis of the control items. Presumably, it would have impacted sample size. Consider adding something on this to fully acknowledge the limitation

• 619 -622: consider rewording for clarity: ‘… we can sometimes actively direct introspective attention to the quality of consciousness as such and we can distinguish possible states by the degree of actually ongoing introspective access to this quality.’ Bold part does not read well and likely to cause confusion

• Observation: 585 (factor 1) and 815 (factor 11) point to the overall mix of phenomenological states reported from respondents in this study. There are a mix of elements that pertain to the more traditionally understood experience of Pure Consciousness directly and others that seem to be as an aside or weaved into the target experience surrounding more commonplace meditative experiences such as being aware of time, effort, duration and agency. These suggest a conceptual point which may be in line with the notion of ‘Glimpsing’ whereby states of pure consciousness for some may be sandwiched by more usual phenomenology, or, as you have suggested, refer to a multidimensional space of possible conscious states, a space which may contain different points, regions or trajectories. Ethically it seems relevant to acknowledge something here – practitioners can be perturbed by states they are taught about in some traditions which often feel illusive and an important part of soteriological endeavours. Acknowledging that in a large sample, experienced meditators have varied experiences including these states and other more banal or commonplace states might provide some instructive benefit for teachers and students

Reviewer #2: This is a well-written and informative paper organised around two key questions: a)

What, if anything, can count as the simplest form of conscious experience? ; b) And is it possible to arrive at a minimal model explanation for conscious experience in neurotypical human beings?

In order to address these questions, the paper proposes to focus on “pure awareness”, using a psychometric approach and a philosophical background theory. The Authors present data from the Minimal Phenomenal Experience (MPE) project, an interdisciplinary initiative aiming at a minimal model explanation of conscious experience. In doing so, the Authors take as a case study the experience of meditation. The overall ambition of this project is to determine whether “pure consciousness” exists and what are its phenomenological markers.

I believe the paper can make a good contribution to the current literature, and I recommend accepting the paper after addressing the Moderate revisions listed below:

- A general theoretical question concerns the very definition of the term “pure consciousness” as awareness of itself: “can a conscious system be exclusively aware of awareness itself ?” line 109. It’s not clear here what is an “awareness itself”. Is it a sort of phenomenon dissociated from the experiencing subject, or is it more like a kantian object (in itself?). What are the motivations behind this definition? This is a key question because the entire paper structure, as well as the survey depends on this definition. P 12 line 172

- Connected with the comment above: p 13 line 197: how many participants failed to understand the term “pure awareness” as the Authors define? Did you keep a record of these participants?

- P 13 line 207 – to which contemplative traditions these advanced practitioners belong to? As this influenced the questionnaire structure indirectly.

- P 20 line 340: meditation is a vast program with important differences in traditions. What is the “other” here refers to?

- P 40 line 678 “pure awareness knows itself timelessly” – I am struggling understanding this phrase. Is pure awareness a sort of homunculus that knows itself intrinsically? Is it epistemic or experiential stance?

- P 41 – line 710 NREM acronym stands for …

- P 42 – I would move the paragraph 735 earlier in order to follow the structure: “traditionally X means Y, but here we mean Z”

- One general worry I have is that the sample size is way to heterogenous and undercontrolled to back up a specific claim. Perhaps the next step should divide the sample in (i) different contemplative practices traditions ; (ii) years of practice; (iii) school of practice; (iii) gender ; (iv) socio-economical background ; (v) history of mental distress and potential disorders;

- One last worry concerns the outlining of positive only aspects of meditative practices. Or, it is well documented that these practices that aim at attaining a certain “pure awareness” can lead to negative experiences and even distress (see Lindahl & Britton 2019). The journey toward the “pure awareness” does not see straightforward and the questionnaire should include the “dark” side of the journey as well.

6. PLOS authors have the option to publish the peer review history of their article (what does this mean?). If published, this will include your full peer review and any attached files.

Reviewer #1: No

Reviewer #2: No

---

## [Decision Letter · Decision Letter 1]

11 Jun 2021

The Minimal Phenomenal Experience questionnaire (MPE-92M): Towards a phenomenological profile of “pure awareness” experiences in meditators

PONE-D-20-34727R1

Dear Dr. Gamma,

We’re pleased to inform you that your manuscript has been judged scientifically suitable for publication and will be formally accepted for publication once it meets all outstanding technical requirements.

Thank you for your patience with the review process. We are sorry it has been such a lengthy wait since submission, but it proved extremely hard to find reviewers.

Kind regards,

Jane Elizabeth Aspell, PhD

Academic Editor

PLOS ONE

Additional Editor Comments (optional):

Reviewers' comments:

Reviewer's Responses to Questions

**Comments to the Author**

1. If the authors have adequately addressed your comments raised in a previous round of review and you feel that this manuscript is now acceptable for publication, you may indicate that here to bypass the “Comments to the Author” section, enter your conflict of interest statement in the “Confidential to Editor” section, and submit your "Accept" recommendation.

Reviewer #1: All comments have been addressed

Reviewer #2: All comments have been addressed

2. Is the manuscript technically sound, and do the data support the conclusions?

Reviewer #1: Yes

Reviewer #2: Yes

3. Has the statistical analysis been performed appropriately and rigorously? 

Reviewer #1: Yes

Reviewer #2: Yes

4. Have the authors made all data underlying the findings in their manuscript fully available?

Reviewer #1: Yes

Reviewer #2: No

5. Is the manuscript presented in an intelligible fashion and written in standard English?

Reviewer #1: Yes

Reviewer #2: Yes

6. Review Comments to the Author

Reviewer #1: Thanks for your detailed responses and ammendments to the paper. Suggestions have been taken on board and changes made that add conceptual clarity, further acknwoledgement of methodological limitations and additional discussion around pure conscious states.

Happy to reccommend this for publication as a very solid and fascinating contribution that should be out there for others to appreciate and learn from.

Reviewer #2: (No Response)

7. PLOS authors have the option to publish the peer review history of their article (what does this mean?). If published, this will include your full peer review and any attached files.

Reviewer #1: No

Reviewer #2: No

---

## [Editor Report · Acceptance letter]

15 Jun 2021

PONE-D-20-34727R1 

The Minimal Phenomenal Experience questionnaire (MPE-92M): Towards a phenomenological profile of “pure awareness” experiences in meditators 

Dear Dr. Gamma:

I'm pleased to inform you that your manuscript has been deemed suitable for publication in PLOS ONE. Congratulations! Your manuscript is now with our production department. 

Kind regards, 

on behalf of

Dr. Jane Elizabeth Aspell 

Academic Editor

PLOS ONE